# PET: Preference Evolution Tracking with LLM-Generated Explainable Distribution

## Abstract

Understanding how user preference evolves over time is a fundamental challenge central to modern digital ecosystems, for which Large Language Models (LLMs) are an increasingly popular approach due to their ability to comprehend the rich semantic context within behavioral data. A common practice is to use LLMs to predict a user's next action by directly generating a ranked list of preferred items. Although effective for short-term prediction, the end-to-end generation paradigm inherently limits personalization. Its opaque decision-making process obscures holistic user profiling and exacerbates popularity bias. To address these limitations, we propose Preference Evolution Tracking (PET), a framework that reframes the task as inferring a dynamic probability distribution over a stable and interpretable lattice of preference clusters. By applying logit-probing and generative classification techniques, PET infers a user's preference as a probability distribution, enabling transparent preference learning. On public benchmarks (Yelp, MovieLens), PET improves ranking quality by up to $40\%$ in NDCG and fairness by $30\%$ in entropy score over direct generation baselines. On a large-scale, real-world dataset from a short-video platform, it excels at ranking long-tail contents, significantly outperforming a SOTA production model by 7 times in the NDCG score. Ultimately, PET transforms the user profile model from direct preference list generation to a transparent distributional preference mapping, paving the way for more explainable, fair, and diverse personalization systems.

## 1 Introduction

Understanding user interests in a transparent and interpretable manner is a central challenge in building trusted and adaptive digital ecosystems. The advent of Large Language Models (LLMs) presents a transformative opportunity in this domain. With their profound ability to comprehend context and semantics within unstructured user behavioral histories – from product reviews to content consumption sequences – LLMs promise to move beyond simple pattern matching to a deeper, more holistic understanding of user intent. The ultimate goal is not merely to predict the next action of users, but to create transparent and dynamic models of user interest that can be trusted, analyzed, and fairly acted upon.

The prevailing approach for capturing user preferences involves leveraging LLMs to directly generate ranked outputs (Wang et al., 2025; 2024a; Ngo & Nguyen, 2024; Yu et al., 2024; Deng et al., 2025), a method proven effective for short-horizon, top-$k$ recommendation tasks. While effective for short-horizon prediction and identifying popular preferences, this direct-to-ranking paradigm leaves important gaps. First, it lacks interpretability: although LLMs encode rich internal states, these models do not expose an explicit or structured representation of the user's overall preferences – making them difficult to audit or reuse in downstream tasks. Second, the focus on a small $k$, combined with popularity bias, often leads to the over-representation of mainstream content – neglecting the long-tail interests that define a user's niche tastes and are critical for achieving holistic, unbiased personalization.

These limitations have significant consequences for algorithmic decision-making in user-oriented applications. The lack of an explicit user model undermines system transparency, making it difficult to govern and audit for fairness and bias. Additionally, the over-reliance on popular items not only distorts model predictions but also reinforces popularity during training – suppressing the model's

ability to learn users' nuanced, long-tail preferences. This feedback loop amplifies the Matthew Effect (i.e., the rich get richer), reinforcing echo chambers that reduce content diversity, entrench mainstream tastes, and marginalize underrepresented creators and user segments (Bakshy et al., 2015; Wu et al., 2024). This motivates a paradigm shift toward transparent, distributional user modeling, leading to our central research question:

*How can we infer personalized evolving preference distribution via LLM – enhancing interpretability, fairness, and diversity?*

To address this problem, we propose Preference Evolution Tracking (PET), a framework that customizes LLMs into probabilistic inference engines for modeling users' evolving preference distributions. By leveraging pre-softmax logits over a stable preference cluster lattice (e.g., categories or tags), PET captures structured relevance signals more effectively than volatile item-level spaces. This approach enhances interpretability and diversity, especially for long-tail preferences. Specifically, PET supports two scalable inference strategies: *Likelihood-based Probing*, which iteratively queries the model for each cluster to build a precise distribution, and the more efficient *Generative Classification*, which extracts all probabilities in a single forward pass. Furthermore, PET employs *Hierarchical Probing* to maintain tractable inference in extreme multi-label scenarios through semantic taxonomy traversal. Applied to domain-aligned LLMs, PET produces portable, interpretable user profiles for long-horizon ranking, fairness auditing, and long-tail preference modeling.

To validate PET's effectiveness, we conduct comprehensive experiments across three diverse datasets with varying cluster complexity: MovieLens, Yelp, and a large-scale dataset from a world-leading short video platform. Our results demonstrate that the distributional approach significantly outperforms direct generation (DG) baselines and state-of-the-art (SOTA) ranking models on public benchmarks. More critically, on the real-world data of the short video platform, our framework shows a unique ability to rank users' niche, long-tail interests, significantly outperforming a SOTA model in production. Notably, our goal is not to replace mature production models on highly popular (head) content. Rather, PET complements them by recovering user-specific long-tail preferences that are typically underserved by popularity-optimized systems. Across all experiments, our methods prove more adept at capturing stable, long-term preferences over transient, short-term interests. Our primary contributions are threefold:

1. **A New Paradigm for User Modeling:** We introduce a novel framework that shifts the focus from optimizing short-term item predictions to generating interpretable, dynamic probability distributions over a stable lattice of human-understandable preference clusters.
2. **A Portable and Dynamic Preference Representation:** PET encodes each user as an interpretable, cluster-level probability distribution over preferences, capturing long-tail interests and remaining portable across downstream tasks such as grouping users with similar preference distributions, tracking preference shifts over time, and conducting fairness audits.
3. **Extensive Empirical Validation:** We validate PET on three datasets (MovieLens, Yelp, short-video platform), achieving $+40\%$ ranking quality (NDCG) and $+30\%$ fairness score (entropy) over *direct generation*, and $+20\%$ NDCG and $-23\%$ in JS-divergence over a SOTA ranking model (Qwen3-Reranker-8B (Zhang et al., 2025)) on public benchmarks, and a $7\times$ improvement in long-tail ranking against a SOTA production-level ranking baseline on a real-world dataset.

## 2  RELATED WORK

**User Preference Modeling.** To deliver effective personalization, user models aims to capture preferences that are both diverse and dynamic, through three key lines of work. *Multi-interest models* represent users via multiple embeddings linked to distinct interests (Shi et al., 2023; Cen et al., 2020; Zhou et al., 2018). *Sequential models*, such as FPMC (Rendle et al., 2010), GRU4Rec (Hidasi et al., 2015), SASRec (Kang & McAuley, 2018), and BERT4Rec (Sun et al., 2019), focus on temporal dynamics to predict the next item in a user's history. More recently, distributional modeling has become central in alignment and fairness, with methods like GDPO for group fairness (Yao et al., 2024) and RLHF-based techniques like DPL and AOT for reward uncertainty (Siththaranjan et al., 2023; Melnyk et al., 2024). While powerful, these alignment-based methods introduce significant practical considerations, as they often require costly preference-labeled datasets and complex, computationally intensive pipelines.

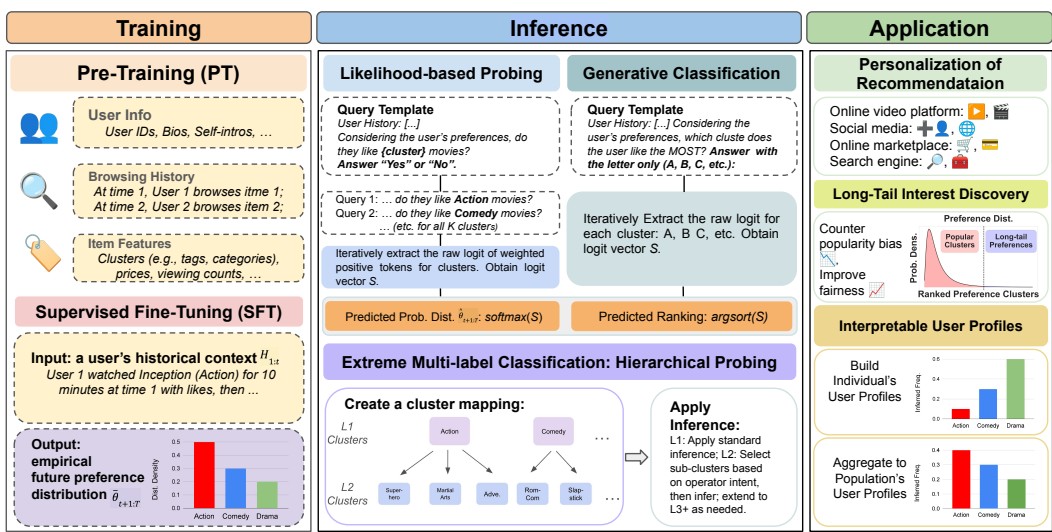

Figure 1: The PET framework pipeline. Left (Training): An LLM is trained on user history to learn preference distributions. Center (Inference): probing methods extracts the predicted preference distribution from the model's internal logits. Right (Application): The transparent distribution is used for downstream tasks: personalized ranking, long-tail discovery, and interpretable user profiling.

**LLMs in Preference Modeling.** Recent work has increasingly applied Large Language Models (LLMs) to the complex task of user preference modeling. Early efforts leveraged LLMs as powerful components for embedding generation (U-BERT; Qiu et al. (2021)) or unified the field under a prompt-based, text-to-text paradigm (P5; Geng et al. (2022)). This led to end-to-end generative models like OneRec (Deng et al., 2025) and (Ngo & Nguyen, 2024) that directly produce ranked outputs or the most relevant items (Wang et al., 2025; 2024a), often incorporating sophisticated alignment modules. In a parallel effort toward explainability, RecGPT uses the LLM to generate descriptive, textual user profiles (Yi et al., 2025). Crucially, the direct generation of ranked lists via LLM – the approach that forms the basis of our primary baseline – has been shown to consistently outperform traditional sequential models (like SASRec (Kang & McAuley, 2018) and BERT4Rec (Sun et al., 2019)) in various settings (Geng et al., 2022; Wu et al., 2024; Zhao et al., 2024; Deng et al., 2025).

Recent approaches also tackle complex scenarios through structured prompting and multi-context modeling, such as LLM4MSR (Wang et al., 2024b), HUM (Bao et al., 2025), and user interest exploration via latent clusters (Wang et al., 2024a). While effective for short-horizon prediction, these methods typically encode a user's state implicitly within the model's parameters, without surfacing an interpretable or temporally grounded preference distribution. Our work is also related to *logit-probing* and *generative classification* techniques for extracting model beliefs from pre-softmax logits (Petroni et al., 2019; Schick & Schütze, 2020; Raffel et al., 2020; Zhou et al., 2024), adapting them to infer evolving user preference distributions from fine-tuned LLMs.

## 3 THEORETICAL AND METHODOLOGICAL FRAMEWORK

In this section, we formalize the problem of inferring dynamic, comprehensive user preferences and introduce the Preference Evolution Tracking (PET) framework. We start by outlining the fundamental components of the proposed framework (see Figure 1).

### 3.1 PROBLEM FORMULATION

We consider a setting in which the user's interaction history up to time $t$ is represented as a sequence $H_{1:t}$. The user's preference at any discrete time $t$ is defined as a probability distribution $\theta_t$ over a

fixed, finite set of $K$ preference clusters $C := \{c_1, \cdots, c_K\}$:

$$\theta_t := (\theta_t(1), \cdots, \theta_t(K)), \text{ with } \theta_t(i) \geq 0 \ \forall i \in [K] \text{ and } \sum_{i \in [K]} \theta_t(i) = 1. \tag{1}$$

We adopt a *latent utility model* (McFadden, 1972; Luce et al., 1959; Train, 2009), a standard formulation in choice modeling, to describe how this true distribution arises. We assume that there exists a vector of unobserved "attractiveness" scores $\boldsymbol{q} := (q_1, \cdots, q_K)$, where $q_i$ represents the true strength of the user's preference for cluster $c_i$. The true preference distribution $\theta_t$ is then generated via a softmax function:

$$\theta_t(i) = \frac{\exp(q_i)}{\sum_{j \in [K]} \exp(q_j)}, \forall i \in [K]. \tag{2}$$

Our primary goal is to learn a function, parameterized by an LLM $\mathcal{M}$, that maps this interaction history to a prediction of the user's future preference distribution, $\hat{\theta}_{t+1:T}$, for a given prediction window $\{t + 1 : T\}$. We choose to model preferences over this stable cluster space rather than the item space, as it is more robust to the high volatility and enormous scale of items in real-world digital ecosystems.

Since this true distribution $\theta_{t+1:T}$ is not directly observable, we construct an empirical proxy for it from the user's interactions in a time window, which we denote as $\bar{\theta}_{t+1:T}$. Formally, for each cluster $c_i \in C$, its probability is the normalized frequency of interactions:

$$\bar{\theta}_{t+1:T}(i) := \frac{\#\{\text{interactions with cluster } c_i \text{ in window}[t + 1, T]\}}{\sum_{j \in [K]} \#\{\text{interactions with cluster } c_j \text{ in window}[t + 1, T]\}}. \tag{3}$$

To ensure reproducibility and simplicity, we use this transparent and normalized frequency of a single action instead of heuristic weighting of different interaction types, as our framework is designed to be agnostic to the specific construction of the ground-truth label. Formally, we aim to minimize the expected divergence between these two distributions:

$$\min_{\hat{\theta}_{t+1:T}} \mathbb{E}_{H_{1:t}, \mathcal{M}} \left[ \mathcal{D} \left( \bar{\theta}_{t+1:T} \parallel \hat{\theta}_{t+1:T} \right) \right], \tag{4}$$

where $\mathcal{D}$ is a distributional loss, such as cross-entropy or KL divergence, as commonly used in LLM post-training. In practice, we do not observe $\theta_{t+1:T}$, so we approximate this population objective by using the empirical proxy $\bar{\theta}_{t+1:T}$ in place of $\theta_{t+1:T}$ in our training data. The ranked list of preferences used for evaluation with metrics like NDCG is then obtained by sorting the elements of $\hat{\theta}_{t+1:T}$ in descending order.

### 3.2 METHODOLOGY

This section outlines the PET framework for learning user preferences, which consists of two core stages: *model alignment* and *preference inference*.

In the *alignment* stage, we adopt a two-phase training pipeline. First, a base LLM is pre-trained (PT) on large-scale domain-specific corpora (e.g., user and item info) to establish foundational knowledge and capture population-level patterns. Second, we align the model to user-level preference prediction via supervised fine-tuning (SFT) on curated interaction sequences. Concretely, from each user's full history we construct SFT examples as pairs (context, label), where the context is an expanding window $H_{1:t}$ and the label is a textual encoding of the empirical future preference distribution $\bar{\theta}_{t+1:T}$.

In practice, we implement SFT as lightweight adapter tuning of the base LLM using LoRA: given a PET prompt and history $H_{1:t}$, the model is trained with standard next-token cross-entropy to generate the target description of $\bar{\theta}_{t+1:T}$. This procedure induces an internal *preference head* that maps histories to logits over clusters, $S(H_{1:t}) \in \mathbb{R}^K$, with $\hat{\theta}_{t+1:T} = \text{softmax}(S(H_{1:t}))$. For our theoretical analysis in Appendix D, we abstract away the textual interface and directly study this induced head as an idealized predictor trained to minimize cross-entropy between the latent distribution $\theta_{t+1:T}$ and $\hat{\theta}_{t+1:T}$. This aligned model forms the basis for all subsequent *preference inference* methods.

We now present two inference methods for preference distribution estimation, which differ primarily in their querying strategy (more detailed design of the prompts and methods are presented in Appendices A and B).

**Method 1: Likelihood-based Probing.** This approach decomposes a multi-class problem into a series of independent binary classifications. The LLM is used as a "prober" or "scorer." For each cluster $c \in C$, we iteratively ask the LLM for inferences and extract the logit score of positive tokens and then apply a softmax function on the score vector to recover the inferred probability distribution (Petroni et al., 2019; Schick & Schütze, 2020).

**Method 2: Generative Classification.** This method uses a single forward pass. A prompt is constructed that frames the task as a multi-choice question, with each cluster mapped to a unique token (e.g., 'A', 'B'). We then read the model's next-token logits for these specific cluster tokens and normalize them via a softmax function to obtain the final probability distribution (Raffel et al., 2020; Chung et al., 2024).

**Extreme multi-label preference inference.** To scale PET to real-world applications involving thousands of fine-grained preference clusters – such as in large video platforms or e-commerce systems – we introduce a two-stage *Hierarchical Probing* strategy. Existing methods face clear limitations in such settings: *Likelihood-based Probing* exhibits linear computational complexity in the number of clusters $K$, becoming prohibitively expensive, while *Generative Classification* often yields unstable and semantically ungrounded logits when forced to assign probabilities to artificial class tokens.

To overcome these issues, we organize clusters into a two-level semantic taxonomy: coarse-grained L1 categories (e.g., Food & Drink, Technology) partition the full L2 label space into interpretable subgroups. This tree-structured approach is also validated in recent extreme multi-label classification work in other tasks (Chen et al., 2023; Wan et al., 2023; Zhou et al., 2024). Inference proceeds in two stages: (1) *L1 Preference Scoping:* estimates marginal probabilities over top-level categories; (2) *Strategic L2 Exploration:* selectively probes child clusters, guided by downstream goals. For example, one may prioritize top-ranked L1 branches for relevance, or intentionally explore long-tail branches for novelty. This decomposition improves both tractability and semantic alignment by guiding the LLM from general categories to relevant fine-grained clusters, enabling scalable and adaptive preference modeling. See the detailed computational complexity analysis in Appendix C.

## 3.3 THEORETICAL GROUNDING AND STRUCTURAL ANALYSIS

This section establishes the theoretical grounding for PET, examining its behavior under varying assumptions of calibration and highlighting its structural advantages over direct generation (DG).

**Idealized regime: perfectly calibrated logits.** Under the latent-softmax model of Section 3, if the PET head is trained in the population limit with cross-entropy, then the induced distribution $\hat{\theta}_{t+1:T}$ recovers the true latent distribution $\theta_{t+1:T}$ almost everywhere (Gneiting & Raftery, 2007; Goodfellow et al., 2016; Blasiok et al., 2023). In this ideal regime, the logits $\boldsymbol{S}$ and latent utilities $\boldsymbol{q}$ differ only by an additive constant, so sorting logits is equivalent to sorting utilities. We show in Appendix D that PET then coincides with the Bayes-optimal ranking, and no decoding-based procedure applied to the same logits can improve any standard order-aware metric (e.g., NDCG, Recall) (Tewari & Bartlett, 2007).

**Imperfect regime: approximated isotonicity.** Real models are not perfectly calibrated, and logits can deviate from the ideal ranking. To model this, we work in an imperfect regime where the PET logits are only *$\varepsilon$-approximately isotonic*: for any pair of clusters $(i, j)$ with $q_i > q_j$, PET misorders the pair (i.e., $S_i < S_j$) with probability at most $\varepsilon$ over training and probing randomness. Let $\mathcal{R}(\pi; \boldsymbol{q})$ denote the number of strictly preferred pairs that are reversed under a ranking $\pi$. We show (Appendix D) that the ranking induced by PET satisfies

$$\mathbb{E}\big[\mathcal{R}(\pi_{\text{PET}}; \boldsymbol{q})\big] \leq \binom{K}{2}\varepsilon,$$

so whenever the PET head is "mostly order-preserving" (small $\varepsilon$), its ranking is provably close to the Bayes-optimal ordering. in Appendix D, we further connect $\varepsilon$ to the training objective by showing

that the excess cross-entropy risk controls the $L_1$ distance between $\theta_{t+1:T}$ and $\hat{\theta}_{t+1:T}$, so improved training directly tightens this regret bound.

**Robustness in the imperfect regime: Structural advantages over direct generation.** In the practical "imperfect" regime, PET exhibits structural advantages over decoding-based DG. While both methods rely on the same base logits $S$, DG is susceptible to three well-documented limitations:

- *Frequency Bias.* Autoregressive LLMs are known to exhibit calibration biases, often favoring high-frequency patterns from pre-training (Zhao et al., 2021; Holtzman et al., 2020). In DG, this popularity bias influences the ranking via next-token probabilities. PET instead normalizes logits over the cluster set, mitigating the effect of raw token frequency when forming the preference distribution.

- *Search Limitations.* DG relies on heuristic search (e.g., beam search) which can exhibit pathological behaviors where higher probability sequences do not necessarily yield better quality (Stahlberg & Byrne, 2019). Crucially, beam search acts as a hard truncation on the long tail: valid niche clusters falling outside the beam are effectively pruned (Wiseman & Rush, 2016). PET computes a global softmax, preserving the relative ordering of the tail without such pruning.

- *Exposure Bias.* DG is autoregressive; early deviations (e.g., selecting a popular but less relevant item) change the context for subsequent predictions (Ranzato et al., 2015; Bengio et al., 2015). PET reads out scores marginally from the history $H_{1:t}$, ensuring that ranking estimates are local and do not compound errors.

Taken together, , these structural properties, along with our theoretical guarantees in Appendix D, help explain our empirical findings: PET's distributional, one-shot ranking remains robust on long-tail tasks, while DG degrades as $K$ and the target list length grow due to truncation and exposure-bias effects layered on top of the same logits.

## 4 EXPERIMENT

### 4.1 EXPERIMENT DESIGN AND SETUP

Our experimental design is crafted to comprehensively evaluate the efficacy of PET's ability in discerning and predicting user preferences across varying granularities and long-term and short-term dynamics. We focus on using two primary inference methodologies for preference prediction and benchmark them against established and SOTA baselines.

**Datasets.** We evaluate PET on three real-world datasets of varying scale and complexity. *Movie-Lens* (Harper & Konstan, 2015): a widely-used dataset for movie recommendations, representing a relatively small and well-defined genre (cluster) space with 19 clusters (i.e., movie genres). In total, we used 200k users with 33 million rating histories associated with 87k movies. *Yelp*: a large-scale dataset with 2 million users and 8 million ratings across 1,311 business categories, providing a significantly higher-dimensional preference space. *A Real-World Short-Video Dataset*: this proprietary dataset is derived from the a real-world short video platform and captures large-scale real-world user interaction behaviors in a highly dynamic environment, reflecting diverse and rich user preference patterns. To ensure privacy, all identifiers are rigorously anonymized. The dataset comprises 50 million implicit feedback records from 16k randomly sampled users, covering 15 million items. Each record contains only anonymized user IDs, item IDs, and interaction types.

**Training and Evaluation Setup.** For the MovieLens and Yelp datasets, where user interactions are relatively sparse, we adopt a temporal 80/20 split at the session level. The first 80% of each user's sessions are used as input for both pretraining (PT) and supervised fine-tuning (SFT), while the remaining 20% is held out for evaluation. For the 20% of the evaluation split data, we measure two targets based on context horizons: (1) encompass the entire lifetime of user behavior for long-term preference prediction, and (2) only hold out the first session for short-term (next-item) prediction. For the denser short-video dataset, we apply a fixed-range protocol across all stages: a 30-day interaction window is used for PT and SFT input, and the subsequent 14-day period is used for evaluation.

**Models and Methods.** We use two families of open-source 8B-parameter language models: *Qwen3-8B* (QwenTeam, 2025) and *DeepSeek-Distill-Llama-8B* (DeepSeek-AI, 2025), selected for their

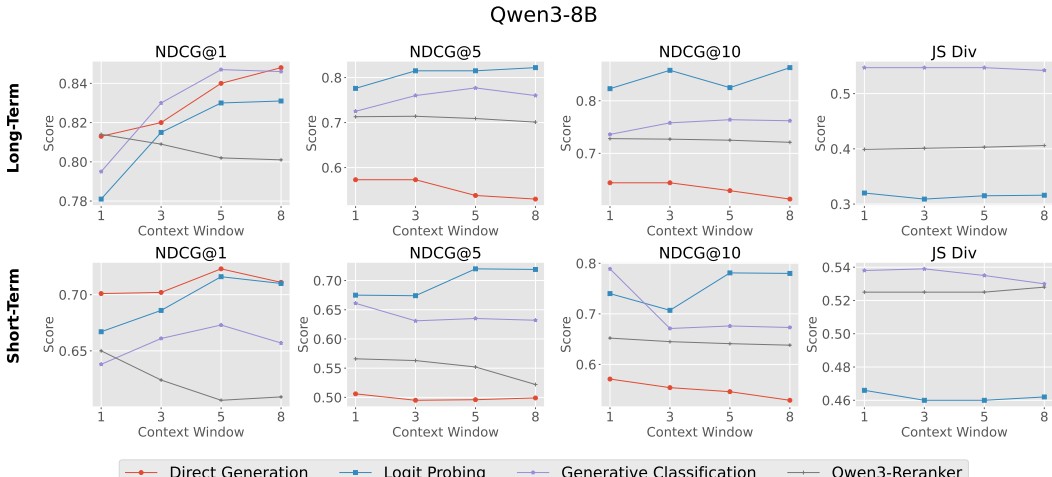

Figure 2: Comprehensive comparison of different methods on MovieLens dataset across a range of context windows (i.e., 1,3,5, and 8 sessions). Prediction windows: long-term and short-term. Metrics: NDCG@$1, 5, 10$ and JS-Divergence. We compare our *Logit Probing* and *Generative Classification* methods against (1) *Direct Generation* benchmark trained (PT+SFT) on a Qwen3-8B base model and (2) a SOTA Qwen3-Reranker-8B ranking model. Note that for the *Direct Generation* baseline, the JS-divergence will not be available. Sample size: 38,434.

strong balance of SOTA performance and computational efficiency. Each model is trained with our PT+SFT (LoRA fine-tune (Hu et al., 2022)) pipeline. This setup supports a comprehensive ablation of training strategies. Our experiments evaluate the primary inference methodologies detailed in Section 3.2: *Likelihood-based Probing* (Algorithm 1) and *Generative Classification (Algorithm 2)*. For the Yelp dataset with large cluster space, we specifically evaluate our *Hierarchical Probing* approach (Algorithm 3). For all methods, the final inferred probability distribution is used to generate a ranked list of clusters for evaluation.

**Baselines.** We compare PET with two strong LLM-based baselines. To isolate the effect of our inference paradigm, the first baseline *Direct Generation* uses the same pre-trained and fine-tuned LLM as PET but is prompted to directly generate a top-k ranked list of preferences without producing a probability distribution (Geng et al., 2022; Deng et al., 2025; Wang et al., 2025; 2024a). As this method does not produce a probability distribution, it is evaluated only on ranking metrics. The second baseline is *Qwen3-Reranker-8B*, a SOTA pre-trained model specifically optimized for ranking tasks (Zhang et al., 2025). We use it out of the box without any additional fine-tuning. Given a user history and a candidate cluster space, it produces both a ranked list and a probability distribution. For the short-video dataset, we compare against the platform's production Transformer-based ranker model.

**Evaluation Metrics.** Model performance is comprehensively evaluated using established metrics compared to long-term (LT) and short-term (ST) ground truths, with a focus on ranking quality and distributional similarity. **Ranking** metrics include NDCG@k, which assesses the quality of ranking order with an emphasis on top positions; Recall@k, measuring the proportion of relevant items captured within the top k results; and Precision@k, evaluating the exactness of top-k recommendations. Distribution similarity is quantified using Jensen-Shannon Divergence (JS-Div), where lower values indicate closer alignment between predicted and actual preference distributions. To quantify **diversity** and **fairness**, we report Global Exposure Entropy Adomavicius & Kwon (2011), defined as the Shannon entropy of the aggregate distribution of clusters recommended across all users at a specific list cutoff $k$. Higher entropy indicates a more equitable distribution of exposure across the preference lattice, mitigating popularity bias and mode collapse. All results are averaged across test samples and reported per context window size as specified in the dataset. (See Appendix E.1 for details.)

4.2 How Effective and Robust is PET across Different Context Horizon?

We begin by evaluating PET on the MovieLens dataset with 19 clusters, aiming to systematically test its effectiveness and robustness across following key dimensions: (1) different input history lengths and (2) prediction horizons (short-term vs. long-term). We vary the input context window from 1 to 8 sessions, and assess performance on both short-term (next-session) and long-term (multi-session) preference prediction.

As shown in Figure 2, the results demonstrate the robustness and effectiveness of PET across a wide range of settings. The performance gains are especially pronounced in full-list metrics (NDCG@5, NDCG@10), where PET's distributional inference enables richer modeling of user preferences beyond top-1 accuracy. For example, with Qwen3-8B, context window = 8, and long-term prediction, *Logit-Probing* achieves NDCG@10 = 0.863, with a +55% improvement over *Direct Generation* and +20% gain over Qwen3-Reranker-8B. It also improves distributional alignment, reducing JS-Div to 0.316, a 22% relative reduction compared to the Qwen3-Reranker-8B's 0.406. While *Direct Generation* is occasionally competitive at NDCG@1, its performance deteriorates as the number of predicted clusters increases, as it might suffers from exposure bias, noted in Section 3.3. Our *Generative Classification* method also shows consistent improvements over *Direct Generation* on Qwen3-8B base model. Though slightly less accurate than *Logit-Probing*, it offers a compelling latency–accuracy trade-off due to its single-pass nature, making it a strong candidate for real-time or resource-constrained environments.

Table 3 presents the *global exposure entropy* results. We observe that the DG baseline suffers from severe mode collapse at low $k$ (e.g., Entropy@1 drops as low as 0.005 on DeepSeek-8B), indicating it defaults to recommending the same few popular genres to nearly all users as the primary choice, i.e., the popularity bias. In contrast, PET (*Logit-Probing*) maintains high entropy even at $k = 1$ (up to 2.307), demonstrating that it successfully surfaces diverse, user-specific interests at the very top of the ranking. While entropy metrics naturally saturate at larger $k$ due to the finite cluster space (19 genres), PET's ability to maintain high diversity at the top – without sacrificing accuracy (as shown by superior NDCG scores) – confirms it effectively mitigates popularity bias where it matters most. (See more experimental results in other settings in Appendix F)

To demonstrate PET's ability to build interpretable profiles, we sampled 125 users and divided their histories into four equal time periods. Using PET's *Likelihood-based Probing*, we inferred a distribution over 19 genres for each user in each period. Aggregating these across users yielded group-level preference distributions that evolve over time – visualized in both a heatmap and 3D ribbon plot (Figure 3). These results show that user preference distributions are not static but migrate in nuanced ways. Dominant genres like *Drama*, *Comedy*, and *Action* exhibit stability across periods, representing structural, long-term user interests. In contrast, genres such as *Adventure*, *Romance*, and *Sci-Fi* demonstrate significant temporal fluctuations – highlighting PET's ability to capture volatile, context-dependent preferences. Notably, genres like *Sci-Fi*, *Fantasy*, and *IMAX* show gradual upward shifts, possibly reflecting emerging or re-surfacing interests. Together, these dynamics validate PET's unique capability not only in preference prediction, but also in generating descriptive, evolving group-level user profiles – an essential feature for diagnosing user behavior shifts, designing cold-start interventions, or understanding preference stability versus exploration.

## 4.3 Can PET Handle Thousands of Clusters?

To evaluate PET's scalability in high-dimensional settings, we use the Yelp dataset with 1,311 fine-grained categories. Since exhaustive *Logit-Probing* becomes intractable at this scale, we adopt a two-stage *Hierarchical Probing* strategy (Algorithm 3). First, we construct a semantic hierarchy that maps all categories into 26 high-level L1 clusters (e.g., "Food & Restaurants," "Health & Medical"). For demonstration, we focus on the high-traffic L1 cluster "Food & Restaurants," further decomposed into 19 L2 sub-clusters (e.g., "Mexican," "Vegan"). During inference, PET first identifies a user's coarse L1 interest, then performs fine-grained ranking within the selected L2 cluster. While operators may target different L1 clusters depending on downstream needs, this experiment provides one representative use case. (See detailed mapping construction in Appendix E.)

Table 1 shows that PET with Hierarchical Probing achieves strong performance across both coarse (L1) and fine-grained (L2) levels. At L1, *Logit-Probing* is unequivocally dominant. It achieves high and robust NDCG scores (e.g., NDCG@1,5,10 are 0.980, 0.879, and 0.915 for Qwen3-8B), demon-

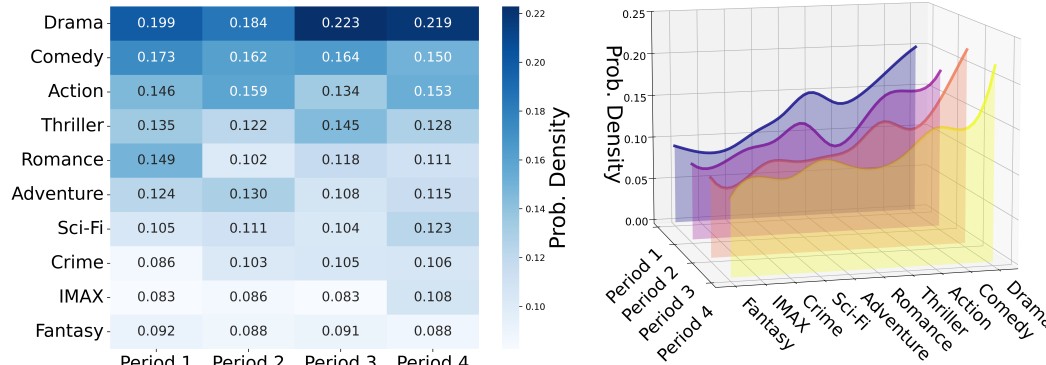

(a) Group-level preference evolution as a probability heatmap

(b) Group-level preference evolution as a bar chart

Figure 3: Evolution of group-level movie genre preferences as probability distribution across four time periods. Dataset: Movielens, 125 users over 4 periods. Method: PET *Likelihood-based Probing* with PT+SFT on Qwen3-8B. Note: due to limited space, we only show top-10 clusters (genres) here.

strating an exceptional ability to identify a user's general interests. In stark contrast, both the *Direct Generation* and *Generative Classification* methods fail at this stage, with performance often near zero. These results reinforce the necessity of iterative probing in high-dimensional settings where broad disambiguation is critical. At L2, where the ranking task becomes more localized, both *Logit-Probing* and *Generative Classification* significantly outperform the *Direct Generation* benchmark. Notably, while *Logit-Probing* remains the top performer for Qwen3-8B (0.666 NDCG@10), *Generative Classification* emerges as the clear winner for the DeepSeek-distill-Llama-8B model (0.574 NDCG@10), suggesting that decoding strategies for fine-grained inference can be model-dependent.

Together, these findings demonstrate that PET, equipped with hierarchical probing, scales effectively to large cluster spaces. The coarse-to-fine process not only preserves tractability but also allows flexible adaptation across model architectures — with *Logit-Probing* excelling in breadth and *Generative Classification* offering a strong, efficient alternative at finer levels of granularity.

| Level | Method | Qwen3-8B | | | DeepSeek-distill-Llama-8B | | |
|---|---|---|---|---|---|---|---|
| | | NDCG@1 | NDCG@5 | NDCG@10 | NDCG@1 | NDCG@5 | NDCG@10 |
| L1 (26 clusters) | Direct Generation | .008 | .139 | .330 | .016 | .308 | .343 |
| | Logit Probing | **.980** | **.879** | **.915** | **.925** | **.856** | **.880** |
| | Generative Classification | .107 | .118 | .163 | .001 | .062 | .119 |
| L2 (19 sub-clusters) | Direct Generation | .298 | .368 | .509 | .200 | .153 | .329 |
| | Logit Probing | **.513** | **.562** | **.666** | .261 | .268 | .363 |
| | Generative Classification | .466 | .349 | .454 | **.684** | **.553** | **.574** |

Table 1: *Hierarchical Probing*'s performance on the high-dimensional Yelp dataset for the long-term prediction task. Results are shown for a context window of 8 using the PT+SFT pipeline. Best NDCG scores for each level are in bold. L1 category contains 26 clusters; L2 category contains 19 clusters under one L1 category of "Food & Restaurants". Alternative taxonomies can be easily integrated depending on platform objectives. Sample Size: 5000.

## 4.4 How Does PET Perform on Real-World Short-Video Platform?

We conclude our empirical study with a large-scale, private short-video dataset from a major content platform, comprising 78 content clusters. This experiment is designed to test our method's efficacy in a real-world environment against a heavily optimized, production-level Transformer-based ranking model. The primary challenge in such an ecosystem is overcoming the inherent feedback loop, where users primarily interact with content they are already shown. Therefore, our evaluation focuses on PET's ability to complement an existing production model that is heavily optimized for popular, high-frequency head content. In contrast, PET is designed to surface niche, long-tail interests that are often suppressed by popularity bias. We aggregate interactions into daily sessions (a 24-hour window), ensuring a denser and more comprehensive view of a user's preferences. All

results use the PT+SFT pipeline with Qwen3-8B. Following industry practice, we use play duration – the total time a user spends watching each content item – as a robust and continuous proxy for engagement. (See more details in Appendix E.3)

Results in Table 2 showcase PET's effectiveness in promoting long-tail interest, diversity, and personalized recommendation. We highlight the setting of 30-day history to 14-day prediction, as it represents the most challenging long-range forecasting task. The production model fails to identify users' long-tail preferences, with an NDCG@20 of just 0.0243 on the long-tail segment. PET with *Logit-Probing* achieves NDCG@20 of 0.1971 on the same tail – a +711% improvement. This demonstrates PET's ability to surface underrepresented, user-specific content that is typically overlooked by popularity-driven production models. Notably, results are consistent for $k \in \{1, 5, 10, 20\}$ with PET achieving more gains compared to the production model when $k$ is small. We also compare across prediction horizons and observe that PET slightly outperforms the baseline more significantly when the forecasting window is longer (14 days vs. 7 days), consistent with our earlier findings at Section 4.2 regarding LLMs' strength in modeling long-term user preferences. Finally, PET's performance remains robust across different context lengths. It maintains stable ranking quality even with limited user history, underscoring its ability to effectively summarize user interests with sparse or noisy data.

Taken together, these findings validate PET as a practical and scalable complement to production systems. enhancing long-tail coverage and personal relevance without sacrificing overall performance. Its ability to enhance diversity and personalization – especially for the most engaged users – marks a valuable addition to real-world recommender systems.

| Context | Prediction | Method | NDCG@1 | NDCG@5 | NDCG@10 | NDCG@20 |
|---------|-----------|--------|--------|--------|---------|---------|
| 14 Days | 7 Days | SOTA Production (Production) | .007 | .009 | .012 | .034 |
| | | Logit-Probing | **.113** | **.124** | **.147** | **.203** |
| | 14 Days | SOTA Transformer (Production) | .004 | .006 | .008 | .024 |
| | | Logit-Probing | **.110** | **.126** | **.150** | **.208** |
| 30 Days | 7 Days | SOTA Transformer (Production) | .007 | .009 | .012 | .034 |
| | | Logit-Probing | **.103** | **.118** | **.138** | **.192** |
| | 14 Days | SOTA Transformer (Production) | .004 | .006 | .008 | .024 |
| | | Logit-Probing | **.108** | **.122** | **.143** | **.197** |

Table 2: Long-tail preference learning performance on the short-video dataset for high-activity users, comparing PET's *Logit Probing* against a SOTA production model across various history (i.e., 14 and 30 days) and prediction windows (i.e., 7 and 14 days). All results are from the PT+SFT pipeline with the Qwen3-8B model.

## 5 CONCLUSION

In this paper, we introduce Preference Evolution Tracking (PET), a framework that casts LLM-based personalization as distributional preference mapping: PET infers a user's cluster-level probability vector via *Likelihood-based Probing* and *Generative Classification*, and scales to large label spaces through *Hierarchical Probing*. Under a mild isotonicity assumption, ranking by PET's inferred probabilities is optimal for standard order-aware metrics. Empirically, PET is robust across datasets and horizons, outperforming strong generative and reranking baselines and substantially improving coverage of long-tail interests on an industrial-scale corpus. Beyond accuracy, PET yields interpretable, temporally evolving user profiles that enable group-level user analysis and preference-shift detection. Future work includes extending PET to model richer user representations by incorporating auxiliary signals such as demographic attributes, behavioral embeddings, and multimodal content. These enhancements aim to enable more personalized, adaptive, and interpretable preference modeling in real-world recommendation systems.

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

## A ALGORITHMS

---

**Algorithm 1** Likelihood-based Probing.

---

**Input:**

$\mathcal{M}$: Large Language Model.

$H_{1:t}$: A user's interaction history from up to time $t$.

$C = \{c_1, \ldots, c_K\}$: The set of $K$ preference clusters.

$T_{\text{probe}}$: A prompt template for probing.

$V_+$: The set of vocabulary token for an affirmative response (e.g., 'Yes', 'Y', 'y', etc.).

$V_-$: The set of vocabulary token for an negative response (e.g., 'No', 'N', 'n', etc.).

$\tau$: temperature parameter.

**Output:** Predicted preferences $\hat{\theta}_{t+1:T}$ over a future period.

1: $S \leftarrow \mathbf{0} \in \mathbb{R}^K$     ▷ Initializations

2: **for** $j \leftarrow 1$ to $K$ **do**

3:     Use $T_{\text{probe}}, H_{1:t}, c_j$ to construct a prompt $p_j$.     ▷ Create prompt for cluster $c_j$

4:     $\mathbf{z}_j \leftarrow \mathcal{M}(p_j)$     ▷ Get logit vector $\mathbf{z}_j \in \mathbb{R}^{|V|}$ for prompt $p_j$

5:     $s_+ \leftarrow \sum_{v \in V_+} (\mathbf{z}_j[v])/|V_+|$     ▷ Calculate mean logit score for all affirmative tokens

6:     $s_- \leftarrow \sum_{v \in V_-} (\mathbf{z}_j[v])/|V_-|$     ▷ Calculate mean logit score for all negative tokens

7:     $S[j] \leftarrow \frac{\exp(s_+)}{\exp(s_+)+\exp(s_-)}$     ▷ Compute 'yes' probability as the cluster's score

8: **end for**

9: $\hat{\theta}_{t+1:T} \leftarrow \text{softmax}(S/\tau)$     ▷ Normalize all scores to get final distribution

---

**Algorithm 2** Generative Classification.

---

**Input:**

$\mathcal{M}$: Large Language Model.

$H_{1:t}$: A user's interaction history from up to time $t$.

$C = \{c_1, \ldots, c_K\}$: The set of $K$ preference clusters.

$T_{\text{gen}}$: A prompt template for generation.

$V_C$: The set of vocabulary token for indexing each cluster.

$\tau$: temperature parameter.

**Output:** Predicted preferences $\hat{\theta}_{t+1:T}$ over a future period.

1: $S \leftarrow \mathbf{0} \in \mathbb{R}^K$     ▷ Initializations

2: Use $T_{\text{gen}}, H_{1:t}, C$ to construct a prompt $p$.

3: $\mathbf{z} \leftarrow \mathcal{M}(p)$.     ▷ Get logit vector $\mathbf{z} \in \mathbb{R}^{|V|}$ for prompt $p$

4: **for** $j \leftarrow 1$ to $K$ **do**

5:     $S[j] \leftarrow \mathbf{z}[v_j]$     ▷ Extract the raw logit for the token of cluster $c_j$

6: **end for**

7: $\hat{\theta}_{t+1:T} \leftarrow \text{softmax}(S/\tau)$     ▷ Normalize all scores to get final distribution

---

## B PROMPT OF INFERENCE

**Prompt for Likelihood-based Probing Algorithm 1**

---

*User History:*

    Time 1: rated "Inception" 5/5 (Action, Sci-Fi);

    Time 2: rated "The Godfather" 5/5 (Crime, Drama);

    Time 3: rated "Toy Story" 4/5 (Animation, Comedy)

*Considering the user's **long-term preferences** from their movie rating history, do they like {GENRE} movies? Answer in "Yes" or "No".*

---

**Algorithm 3** Hierarchical Probing with Strategic Exploration.

**Input:**
    $\mathcal{M}$: Large Language Model.
    $H_{1:t}$: A user's interaction history.
    $C_{L1} = \{c_{1,L1}, \ldots, c_{K1,L1}\}$: The set of $K1$ L1 clusters.
    $C_{L2} = \{c_{1,L2}, \ldots, c_{K2,L2}\}$: The set of $K2$ L2 clusters.
    $\text{map}(c_{j,L1})$: Function mapping an L1 cluster to its subset of L2 children.
**Output:** Predicted preferences $\hat{\theta}$ over the L2 clusters $C_{L2}$.

                                                                    ▷ **Step 1: L1 Preference Scoping**
1: Use $H_{1:t}$ and $C_{L1}$ to get L1 scores $S_{L1}$ via "Likelihood-based Probing" (Algorithm 1).
2: $P_{L1} \leftarrow \text{softmax}(S_{L1})$                                ▷ Normalize L1 scores to get probabilities.

                                                        ▷ **Step 2: Strategic L2 Exploration**
3: $C_{L1,\text{selected}} \leftarrow \text{SelectBranches}(C_{L1}, P_{L1}, \text{strategy})$       ▷ Select branches based on the goal.
4: $S_{L2} \leftarrow \mathbf{0} \in \mathbb{R}^K$                               ▷ Initialize final scores for all K L2 clusters.
5: **for** each selected cluster $c_{j,L1}$ in $C_{L1,\text{selected}}$ **do**
6:      $C_{L2,\text{subset}} \leftarrow \text{map}(c_{j,L1})$                  ▷ Get L2 children for this L1 parent.
7:      Construct a conditional prompt for $C_{L2,\text{subset}}$ (e.g., "Given interest in $c_{j,L1}$, ...").
8:      Use the conditional prompt to get scores $S_{L2,\text{subset}}$ via "Likelihood-based Probing".
9:      $P_{L2|L1} \leftarrow \text{softmax}(S_{L2,\text{subset}})$          ▷ Get conditional probabilities $P(C_{L2}|c_{j,L1})$.
10:      **for** each cluster $c_k$ in $C_{L2,\text{subset}}$ **do**
11:          $S_{L2}[k] \leftarrow P_{L2|L1}[c_k] \times P_{L1}[j]$       ▷ Apply chain rule for joint probability.
12:      **end for**
13: **end for**

                                                            ▷ **Final Normalization**
14: $\hat{\theta} \leftarrow \text{softmax}(S_{L2}/\tau)$        ▷ Normalize all computed scores to get the final L2 distribution.

**Prompt for Generative Classification Algorithm 2**

> *User History:*
>      Time 1: rated "Inception" 5/5 (Action, Sci-Fi);
>      Time 2: rated "The Godfather" 5/5 (Crime, Drama);
>      Time 3: rated "Toy Story" 4/5 (Animation, Comedy)
>
> *Considering the user's **long-term preferences** from their movie rating history, which genre do they like MOST? Answer with the letter only (A, B, C, etc.):*

**Prompt for Direct Generation (top-1)**

> *User History:*
>      Time 1: rated "Inception" 5/5 (Action, Sci-Fi);
>      Time 2: rated "The Godfather" 5/5 (Crime, Drama);
>      Time 3: rated "Toy Story" 4/5 (Animation, Comedy)
>
> *Question:* Based on the user's **long-term preferences** from their entire history, tell me the cluster they like the MOST. Answer with the letter only (A, B, C, etc.):

**Prompt for Direct Generation (top-3)**

## C   COMPUTATIONAL COMPLEXITY ANALYSIS

**Notation.** Let $K$ be the number of preference clusters and $T$ the prompt length (tokens). Let $C(\mathcal{M}, T)$ denote the cost of a single forward pass of the language model $\mathcal{M}$ on a prompt of length $T$ (the dominant compute). Each cluster $c$ is associated with a fixed "verbalizer" (token or short phrase) used to read its score from next-token logits.

### COMPLEXITY OF PET METHODS

**Likelihood-based Probing Algorithm 1.** Our implementation is **iterative**: we issue a focused prompt per cluster, effectively scoring clusters independently. The total cost scales linearly with the number of probes:

$$\text{Cost} \approx O\big(K \cdot C(\mathcal{M}, T)\big).$$

**Generative Classification Algorithm 1.** This method is **single-shot**: one prompt, one forward pass, then read the next-token logits for the $K$ verbalizers and normalize. The cost is constant with respect to $K$:

$$\text{Cost} \approx O\big(C(\mathcal{M}, T)\big).$$

**Hierarchical Probing Algorithm 3.** This two-stage method targets large $K$. Let $K_1$ be the number of coarse L1 clusters, $B$ the number of L1 branches explored, and $K_{2,\text{avg}}$ the average number of L2 sub-clusters per explored branch. Using iterative probing at both levels, the total cost is the sum of L1 scoping and L2 exploration:

$$\text{Cost} \approx O\big(K_1 \cdot C(\mathcal{M}, T_{L1})\big) + O\big(B \cdot K_{2,\text{avg}} \cdot C(\mathcal{M}, T_{L2})\big).$$

Since typically $K_1 \ll K$ and $B \cdot K_{2,\text{avg}} \ll K$, this is substantially cheaper than flat iterative probing $O\big(K \cdot C(\mathcal{M}, T)\big)$.

### COMPLEXITY OF BASELINES (FOR COMPARISON)

**Direct Generation (top-$k$ list).** One prompt with autoregressive decoding of approximately $O(k)$ output tokens:

$$\text{Cost} \approx O\big(C(\mathcal{M}, T) + k\big).$$

If re-prompted separately for multiple cutoffs (e.g., $k = 1, 5, 10$), end-to-end latency increases linearly with the number of cutoffs.

**Cross-encoder Reranker.** Score each (history, cluster) pair independently:

$$\text{Cost} \approx O\big(K \cdot C(\mathcal{M}, T)\big).$$

### PRACTICAL NOTES

- **KV cache reuse.** For iterative methods, reusing the encoded history reduces repeated compute; most marginal cost is in the final token(s) where logits are read.
- **Batching.** Batching improves throughput for iterative probing and reranking, but tail latency still scales with the number of batches/probes.
- **Multi-$k$ evaluation.** PET's distributional methods (Generative Classification; Hierarchical with small $B$) yield a full distribution in the same pass(es), so reporting multiple $k$ does not require re-decoding.

# D   EXTRA THEORIES AND PROOFS

For the theoretical analysis, we consider an idealized *preference mapping* $g_\phi : H_{1:t} \to \mathbb{R}^K$ that produces logits $\boldsymbol{S}(H_{1:t}) = g_\phi(H_{1:t})$ and an induced distribution $\hat{\theta}_{t+1:T}(H_{1:t}) = \mathrm{softmax}(\boldsymbol{S}(H_{1:t}))$. We assume that $\phi$ is chosen to minimize the *population* cross-entropy between the latent-softmax model and the predictor:

$$\mathcal{L}_{\mathrm{CE}}(\phi) = \mathbb{E}_{H_{1:t}}\big[\mathrm{CE}\big(\theta_{t+1:T}, \hat{\theta}_{t+1:T}\big)\big]. \tag{5}$$

Training on empirical labels $\bar{\theta}_{t+1:T}$ can be viewed as a finite-sample approximation to this population objective. For the theoretical analysis in this appendix, we work in an *idealized population setting*. We posit a latent preference distribution $\theta(H_{1:t}) = \mathrm{softmax}(\mathbf{q}(H_{1:t}))$ and view $\bar{\theta}_{t+1:T}$ as a finite-sample Monte Carlo estimate of $\theta(H_{1:t})$ (i.e., $\bar{\theta}_{t+1:T} \approx \theta(H_{1:t})$ as the number of interactions grows). Consequently, the empirical objective with $\bar{\theta}$ approximates the population objective with $\theta$.

## D.1   PRELIMINARIES: IDEALIZED BAYES-OPTIMALITY

We first establish the behavior of the model in an idealized population limit. Let $\boldsymbol{S} \in \mathbb{R}^K$ be the model's logits, produced by PET's probing methods (i.e., Algorithms 1 and 2). The model predicts $\hat{\theta} = \mathrm{softmax}(\mathbf{S})$, and we train by minimizing cross-entropy between $\theta$ and $\hat{\theta}$. This follows the standard view of cross-entropy (log-loss) as a *strictly proper scoring rule*: at the population optimum, the predictor's output distribution coincides with the true conditional distribution (Gneiting & Raftery, 2007; Goodfellow et al., 2016; Blasiok et al., 2023). In multiclass settings, this implies Bayes-consistency of the induced classifier (Tewari & Bartlett, 2007).

**Proposition 1** (Isotonicity at convergence). *Since cross-entropy is a strictly proper scoring rule (Gneiting & Raftery, 2007; Blasiok et al., 2023), the global minimum of the population risk is achieved if and only if* $\mathrm{softmax}(\boldsymbol{S}) = \mathrm{softmax}(\boldsymbol{q})$. *Equivalently, in the latent-softmax model* $\theta = \mathrm{softmax}(\boldsymbol{q})$, *the Bayes-optimal predictor recovers the true conditional distribution (Tewari & Bartlett, 2007; Goodfellow et al., 2016). By injectivity of softmax up to an additive constant, this implies* $\boldsymbol{S} = \boldsymbol{q} + c\mathbf{1}$ *for some scalar c. Thus, in the idealized limit, the model's logits are perfectly order-preserving (isotonic) with respect to latent preferences.*

*Proof.* Let the true latent preference distribution be $\theta = \mathrm{softmax}(\boldsymbol{q})$ and the predicted distribution be $\hat{\theta} = \mathrm{softmax}(\boldsymbol{S})$. The expected population cross-entropy risk is defined as:

$$\mathcal{R} = \mathbb{E}_H\left[-\sum_{k=1}^K \theta_k \log \hat{\theta}_k\right]. \tag{6}$$

By adding and subtracting the entropy of the true distribution $H(\theta) = -\sum \theta_k \log \theta_k$, we can rewrite the risk as:

$$\mathcal{R} = \mathbb{E}_H\left[H(\theta) + D_{\mathrm{KL}}(\theta \| \hat{\theta})\right], \tag{7}$$

where $D_{\mathrm{KL}}$ is the Kullback-Leibler divergence. A fundamental property of proper scoring rules (Gibbs' inequality) states that $D_{\mathrm{KL}}(\theta \| \hat{\theta}) \geq 0$, with equality holding if and only if $\hat{\theta} = \theta$ almost everywhere. Therefore, the global minimum is achieved uniquely when $\mathrm{softmax}(\boldsymbol{S}) = \mathrm{softmax}(\boldsymbol{q})$.

The softmax function is invariant to constant shifts, meaning $\mathrm{softmax}(\boldsymbol{x}) = \mathrm{softmax}(\boldsymbol{y})$ implies $\boldsymbol{x} = \boldsymbol{y} + c\mathbf{1}$ for some scalar $c \in \mathbb{R}$. Consequently, at the global minimum, $\boldsymbol{S} = \boldsymbol{q} + c\mathbf{1}$.

Finally, since translation by a scalar $c$ does not alter the relative ordering of elements, for any pair of clusters $i, j$:

$$S_i > S_j \iff (q_i + c) > (q_j + c) \iff q_i > q_j. \tag{8}$$

This confirms that in the idealized limit, the logits $\boldsymbol{S}$ are perfectly isotonic with the latent utility $\boldsymbol{q}$. $\qquad\square$

In the language-modeling context, this corresponds to the standard interpretation of cross-entropy training as maximum-likelihood estimation of the next-token distribution and an upper bound on the true entropy rate (Braverman et al., 2020). Our analysis applies this Bayes-optimal perspective to the cluster-distribution predictor used by PET.

## D.2   ROBUSTNESS: THE $\varepsilon$-APPROXIMATE REGIME

Proposition theorem 1 characterizes an ideal Bayes-optimal limit where $\boldsymbol{S}$ is exactly isotonic with $\boldsymbol{q}$. In practice, models are only approximately calibrated (Blasiok et al., 2023; Braverman et al., 2020). We capture this via a probabilistic pairwise condition.

**Assumption 1** ($\varepsilon$-Approximate Pairwise Isotonicity). *For a given user and time window, let $\boldsymbol{q} \in \mathbb{R}^K$ be latent scores and $\boldsymbol{S} \in \mathbb{R}^K$ the logits produced by PET. We say $\boldsymbol{S}$ is $\varepsilon$-approximately pairwise isotonic w.r.t. $\boldsymbol{q}$ if, for all $i, j$ with $q_i > q_j$,*

$$\mathbb{P}\Big(S_i < S_j\Big) \;\leq\; \varepsilon, \tag{9}$$

*where the probability is over randomness in training, histories, and probing. That is, each strictly preferred pair is misordered with probability at most $\varepsilon$.*

We measure ranking quality by the number of misordered pairs relative to the Bayes-optimal ranking.

**Definition 1** (Pairwise ranking regret). Let $\pi^*$ be the permutation that sorts clusters by decreasing $q_i$, and let $\pi$ be any permutation of $\{1, \ldots, K\}$. Define

$$\mathcal{R}(\pi; \boldsymbol{q}) = \sum_{\substack{i,j \in [K] \\ q_i > q_j}} \mathbb{I}\big(\pi^{-1}(i) > \pi^{-1}(j)\big), \tag{10}$$

where $\pi^{-1}(i)$ denotes the position (rank) of cluster $i$ under the permutation $\pi$ (smaller is better). Thus, $\mathcal{R}(\pi; \boldsymbol{q})$ counts the number of strictly preferred pairs $(i, j)$ that are reversed under $\pi$.

Let $\pi_{\mathrm{PET}}$ be the permutation that sorts $\boldsymbol{S}$ in descending order. The following bound is immediate.

**Theorem 2** (Approximate optimality of PET). *Under Assumption 1, the expected pairwise regret of $\pi_{\mathrm{PET}}$ satisfies*

$$\mathbb{E}\big[\mathcal{R}(\pi_{\mathrm{PET}}; \boldsymbol{q})\big] \;\leq\; \binom{K}{2}\varepsilon = O(K^2\varepsilon), \tag{11}$$

*where the expectation is over the randomness in $\boldsymbol{S}$.*

*Proof.* For $q_i > q_j$, define $E_{ij} = \{\pi_{\mathrm{PET}}^{-1}(i) > \pi_{\mathrm{PET}}^{-1}(j)\} = \{S_i < S_j\}$. By Assumption 1, $\mathbb{P}(E_{ij}) \leq \varepsilon$. By Definition 1, $\mathcal{R}(\pi_{\mathrm{PET}}; \boldsymbol{q}) = \sum_{q_i > q_j} \mathbb{I}(E_{ij})$. Taking expectations and using linearity,

$$\mathbb{E}[\mathcal{R}(\pi_{\mathrm{PET}}; \boldsymbol{q})] = \sum_{q_i > q_j} \mathbb{P}(E_{ij}) \leq \sum_{q_i > q_j} \varepsilon \leq \binom{K}{2}\varepsilon. \tag{12}$$

$\square$

Thus, if PET is "mostly isotonic" in the sense of Assumption 1 (small $\varepsilon$), its ranking is close to Bayes-optimal: only $O(K^2\varepsilon)$ pairs are misordered in expectation. In the limit $\varepsilon \to 0$, we recover the exact isotonicity of Proposition 1.

## D.3   FROM CROSS-ENTROPY TO THE $\varepsilon$ REGIME

The previous result treats $\varepsilon$ as a behavioral property of the PET head. We now connect $\varepsilon$ to the training objective $\mathcal{L}_{\mathrm{CE}}(\phi)$.

Let $\mathcal{L}_{\mathrm{CE}}^\star = \mathcal{L}_{\mathrm{CE}}(\theta_{t+1:T})$ denote the Bayes-optimal risk, achieved when $\hat{\theta}_{t+1:T} = \theta_{t+1:T}$, and define the *excess cross-entropy risk*

$$\Delta_{\mathrm{CE}} = \mathcal{L}_{\mathrm{CE}}(\phi) - \mathcal{L}_{\mathrm{CE}}^\star = \mathbb{E}_{H_{1:t}}\big[\mathrm{KL}\big(\theta_{t+1:T}(H_{1:t}) \,\|\, \hat{\theta}_{t+1:T}(H_{1:t})\big)\big].$$

By Pinsker's inequality and Jensen's inequality, we obtain:

**Lemma 1** ($\mathrm{L}_1$ error from excess cross-entropy). *The expected $L_1$ distance between the latent and predicted preference distributions is bounded by*

$$\mathbb{E}_{H_{1:t}}\big[\|\theta_{t+1:T}(H_{1:t}) - \hat{\theta}_{t+1:T}(H_{1:t})\|_1\big] \;\leq\; \sqrt{2\,\Delta_{\mathrm{CE}}}.$$

*Proof.* Recall that the excess cross-entropy risk is exactly the expected Kullback-Leibler divergence between the true distribution $\theta$ and the predicted distribution $\hat{\theta}$:

$$\Delta_{\text{CE}} = \mathbb{E}_{H_{1:t}} \left[ D_{\text{KL}}(\theta \| \hat{\theta}) \right]. \tag{13}$$

For any specific history $H_{1:t}$, Pinsker's inequality bounds the $L_1$ distance by the KL divergence:

$$\|\theta - \hat{\theta}\|_1 \leq \sqrt{2 \, D_{\text{KL}}(\theta \| \hat{\theta})}. \tag{14}$$

Taking the expectation over histories $H_{1:t}$ on both sides:

$$\mathbb{E}\left[ \|\theta - \hat{\theta}\|_1 \right] \leq \mathbb{E}\left[ \sqrt{2 \, D_{\text{KL}}(\theta \| \hat{\theta})} \right]. \tag{15}$$

Since the square root function $f(x) = \sqrt{x}$ is concave, Jensen's inequality implies that $\mathbb{E}[\sqrt{X}] \leq \sqrt{\mathbb{E}[X]}$. Applying this to the right-hand side:

$$\mathbb{E}\left[ \sqrt{2 \, D_{\text{KL}}(\theta \| \hat{\theta})} \right] \leq \sqrt{2 \, \mathbb{E}\left[ D_{\text{KL}}(\theta \| \hat{\theta}) \right]} = \sqrt{2 \, \Delta_{\text{CE}}}. \tag{16}$$

Combining these steps yields the lemma. $\qquad\square$

As $\Delta_{\text{CE}} \to 0$ (e.g., with more data, capacity, and optimization), the predicted distributions $\hat{\theta}_{t+1:T}$ converge to $\theta_{t+1:T}$ in expected $L_1$ distance. Intuitively, if $\hat{\theta}_{t+1:T}$ is close to $\theta_{t+1:T}$, then it is unlikely to significantly distort the ordering between clusters with a clear preference gap. Formalizing $\varepsilon$ as an explicit function of $\Delta_{\text{CE}}$ for *all* pairs would require additional global margin assumptions on $\theta_{t+1:T}$, which are often unrealistic in long-tail regimes. Instead, we interpret $\varepsilon$ as capturing residual misorderings after training: Lemma 1 shows that cross-entropy minimization drives $\hat{\theta}_{t+1:T}$ toward $\theta_{t+1:T}$, thereby pushing the model into a small-$\varepsilon$ regime for most histories and clearly preferred pairs.

## D.3 PET VS. DECODING-BASED DIRECT GENERATION

We now compare PET to *decoding-based direct generation* (DG), which uses the same logits $\boldsymbol{S}$ but produces a ranked list via an autoregressive decoding algorithm (e.g., greedy, beam, sampling).

Let $\pi^* \in \Pi_K$ be the permutation that sorts clusters by decreasing $\boldsymbol{q}$, and let $\pi_{\text{PET}}$ be the permutation that sorts $\boldsymbol{S}$. We model any decoding-based ranking as follows.

**Definition 2** (Decoding-based ranking). A decoding-based ranking procedure is a (possibly randomized) algorithm that maps logits $\boldsymbol{S} \in \mathbb{R}^K$ to a permutation of clusters. We denote by

$$\pi_{\text{DG}}(\boldsymbol{S}) \in \Pi_K$$

the (random) permutation produced by such a decoder when run on $\boldsymbol{S}$. The randomness here comes from any sampling or tie-breaking used by the decoding algorithm (e.g., top-$p$ sampling, temperature, stochastic beam search). We write $\mathbb{E}_{\text{DG}}[\cdot]$ for expectation with respect to this decoder randomness, conditioned on the logits $\boldsymbol{S}$.

We measure quality using a generic order-aware loss.

**Assumption 2** (Order-aware ranking loss). *Let $\mathcal{L} : \Pi_K \times \mathbb{R}^K \to \mathbb{R}$ be a loss such that, for any fixed $\boldsymbol{q}$:*

1. *$\mathcal{L}(\pi^*; \boldsymbol{q}) \leq \mathcal{L}(\pi; \boldsymbol{q})$ for all $\pi \in \Pi_K$;*

2. *for almost all $\boldsymbol{q}$ (w.r.t. any continuous distribution), the minimizer is unique: $\mathcal{L}(\pi^*; \boldsymbol{q}) < \mathcal{L}(\pi; \boldsymbol{q})$ for all $\pi \neq \pi^*$.*

*This is satisfied, e.g., if L is the negative of NDCG/Recall/Precision.*

In the Bayes-optimal regime of Proposition 1, $\boldsymbol{S} = \boldsymbol{q} + c\mathbf{1}$, PET recovers $\pi^*$ by simple sorting. We show that no decoding procedure can do better on top of these logits.

**Theorem 3** (Dominance of PET over decoding-based generation). *Fix $q \in \mathbb{R}^K$ and assume $S = q + c\mathbf{1}$ for some c. Let $\pi_{\mathrm{PET}} = \mathrm{argsort}\,S$ and let $\pi_{\mathrm{DG}}(S) \in \Pi_K$ be the (possibly random) permutation produced by any decoding-based procedure when run on logits $S$. Under Assumption 2,*

$$\mathbb{E}_{\mathrm{DG}}\big[\mathcal{L}(\pi_{\mathrm{DG}}(S); q)\big] \geq \mathcal{L}(\pi_{\mathrm{PET}}; q), \tag{17}$$

*where the expectation is over the decoder's internal randomness. Moreover, if $\mathcal{L}(\cdot; q)$ has a unique minimizer $\pi^*$ and the decoder outputs any $\pi \neq \pi^*$ with nonzero probability, then the inequality is strict.*

*Proof.* Since $S = q + c\mathbf{1}$, sorting $S$ is equivalent to sorting $q$: $\pi_{\mathrm{PET}} = \mathrm{argsort}\,S = \mathrm{argsort}\,q = \pi^*$. By Assumption 2, $L(\pi^*; q) \leq L(\pi; q)$ for all $\pi \in \Pi_K$.

For any fixed realization of the decoder's randomness, the decoder outputs some permutation $\pi_{\mathrm{DG}}(S) \in \Pi_K$, and thus

$$\mathcal{L}(\pi_{\mathrm{DG}}(S); q) \geq \mathcal{L}(\pi^*; q) = \mathcal{L}(\pi_{\mathrm{PET}}; q). \tag{18}$$

Taking expectation over the decoder's randomness gives

$$\mathbb{E}_{\mathrm{DG}}\big[\mathcal{L}(\pi_{\mathrm{DG}}(S); q)\big] \geq \mathcal{L}(\pi_{\mathrm{PET}}; q), \tag{19}$$

since $\mathcal{L}(\pi_{\mathrm{PET}}; q)$ is deterministic.

If $\mathcal{L}(\cdot; q)$ has a unique minimizer $\pi^*$, then whenever $\mathcal{L}(\pi_{\mathrm{DG}}(S); q) = \mathcal{L}(\pi^*; q)$ we must have $\pi_{\mathrm{DG}}(S) = \pi^*$. Thus equality in expectation can hold only if the decoder returns $\pi^*$ almost surely. If instead the decoder outputs some $\pi \neq \pi^*$ with nonzero probability, then on that event $\mathcal{L}(\pi_{\mathrm{DG}}(S); q) > \mathcal{L}(\pi^*; q)$, which makes the inequality strict. $\square$

Theorem 3 is an idealized statement: when logits are perfectly calibrated ($S = q + c\mathbf{1}$), PET's simple logit-sorting is Bayes-optimal, and any decoding-based mapping on top of the same logits can at best match, but not improve, the induced ranking. In practice, PET operates in the $\varepsilon$-approximate regime of Assumption 1, where its regret is controlled by Theorem 2, while decoding-based methods incur additional errors from exposure bias, truncation, and search heuristics.

**Remark 1** (Risk Decomposition: PET vs. Direct Generation). Theorem 3 characterizes an idealized regime where logits are perfectly calibrated ($S \approx q$). In the practical regime, Direct Generation (DG) incurs additional error sources that PET avoids due to its distributional nature. We formally conceptualize the ranking risk of DG as:

$$\mathrm{Risk}(\mathrm{DG}) \approx \mathrm{Risk}(\mathrm{PET}) + \Delta_{\mathrm{freq}} + \Delta_{\mathrm{trunc}} + \Delta_{\mathrm{exp}}. \tag{20}$$

Here, $\mathrm{Risk}(\mathrm{PET})$ is controlled by the approximate isotonicity parameter $\varepsilon$ (Theorem 2), while the additive penalty terms represent the structural limitations identified in Section 3.3:

1. $\Delta_{\mathrm{freq}}$ (Frequency Bias): Represents the divergence caused by the autoregressive model's tendency to over-produce high-frequency tokens from pre-training (Zhao et al., 2021; Holtzman et al., 2020), which PET mitigates via closed-set normalization.

2. $\Delta_{\mathrm{trunc}}$ (Search/Truncation Error): Represents the expected utility loss from heuristic search (e.g., beam search), which imposes a hard truncation on valid long-tail clusters falling outside the beam (Wiseman & Rush, 2016; Stahlberg & Byrne, 2019).

3. $\Delta_{\mathrm{exp}}$ (Exposure Bias): Represents the error accumulated due to context distribution shift during autoregressive decoding (Ranzato et al., 2015).

This decomposition theoretically grounds our empirical observation that PET is significantly more robust in long-tail ranking tasks.

# E DETAILED EXPERIMENTAL DESIGN

## E.1 SETUPS

**Global Exposure Entropy.** Let $\mathcal{U}$ be the set of evaluation users and $\mathcal{C}$ the cluster space (e.g., genres). For a given method and cutoff $k$, let $R_{u,k} \subseteq \mathcal{C}$ denote the set of clusters appearing in the top-$k$ list

for user $u \in \mathcal{U}$. We define the global exposure count for cluster $c \in \mathcal{C}$ as

$$n_c(k) = \sum_{u \in \mathcal{U}} \mathbb{I}(c \in R_{u,k}), \tag{21}$$

and the corresponding normalized exposure distribution

$$P_{\exp,k}(c) = \frac{n_c(k)}{\sum_{c' \in \mathcal{C}} n_{c'}(k)}. \tag{22}$$

The *Global Exposure Entropy* at cutoff $k$ is then

$$\text{Entropy@}k = H_k := -\sum_{c \in \mathcal{C}} P_{\exp,k}(c) \, \log_2 P_{\exp,k}(c). \tag{23}$$

Higher values of $H_k$ indicate that exposure is distributed more equitably across clusters, while lower values indicate concentration on a small subset of clusters (mode collapse).

### E.2 YELP DATASET

On the Yelp dataset, we evaluate PET in a high-dimensional setting with 1,311 fine-grained categories in the Yelp dataset, designed to test the scalability of our approach. In this regime, *Logit Probing* becomes prohibitively expensive due to the large output space. To address this, we adopt Hierarchical Probing (Algorithm 3) that decomposes the task into two levels: a coarse-grained cluster identification followed by fine-grained ranking within the selected scope.

We construct a hierarchical mapping that maps 1,311 categories into 26 high-level L1 clusters (e.g., "Food & Restaurants", "Health & Medical"). For a high-traffic category like "Food & Restaurants," we then define a corresponding L2 space with 19 sub-clusters (e.g., "Mexican," "Vegan") under L1's "Food & Restaurants". In inference, PET first identifies the user's broad L1 interests and then "zooms in" to perform a nuanced L2 ranking within the top-predicted L1 category. While operators may freely select which high-level categories to probe based on application needs, this experiment offers one representative setup on the Yelp dataset.

**Hierarchical Probing Mapping.** We ask Gemini-2.5 Pro and GPT-5 to produce this mapping with the prompt:

---

*You are an expert in ontology construction, taxonomy design, and user preference modeling, with extensive experience in building interpretable, semantically meaningful cluster hierarchies for preference learning.*

*I will provide you a list of raw user preference clusters, each with an associated weight or frequency representing its importance or user interaction volume (e.g., number of views).*

> Bicycles: number of reviews xxx, average ratings xxx;
> Massage: number of reviews xxx, average ratings xxx;
> ...

*Your task is to:*

1. Group these raw clusters into high-level categories (Level-1) that are semantically meaningful and interpretable.

2. Assign each raw cluster to one and only one Level-1 category, forming a two-level hierarchy.

3. Return the result as a JSON object, where the keys are Level-1 category names and the values are lists of Level-2 clusters (i.e., raw cluster names).

*You should aim for:*

1. 20 to 26 Level-1 categories total.

2. Coherent semantics within each group.

---

3. Human-readable, concise names for each Level-1 category (e.g., "Food & Drink",
"Fitness", "Beauty").

With the mapping, of which we verified that Level-1 categories are well-balanced, with no single group dominating the total frequency or collapsing into overly broad labels:

*L1 Categories*:

Active Life & Fitness, Active Life, Sports & Recreation, Arts, Entertainment & Events, Automotive, Beauty & Spas, Cannabis Services, Community & Government, Education, Event Planning & Services, Farms & Ranches, Food & Restaurants, Health & Medical, Home & Public Services, Home Maintenance, Hotels & Travel, Internet & Communications, Local & Public Services, Nightlife & Bars, Personal Services, Pets, Professional & Financial Services, Real Estate, Religious & Community, Shopping & Retail, Specialty Shops, Specialty Vehicles

*L2 Categories*:
Bicycles, Bike Shop, Active Life, Aerial Fitness, Airsoft, Amateur Sports Teams, Archery, Badminton, Barre Classes, Baseball Fields, Basketball Courts, Beach Bars, Beaches, Bicycle Paths, Bike Parking, Bike Sharing, Bike tours, Bikes, Bocce Ball, Boot Camps, Boxing, Brazilian Jiu-jitsu, ...

### E.3 REAL-WORLD SHORT-VIDEO DATASET

On the short-video dataset, we select the play duration as the proxy metric to represent the user interests. Compared to sparse binary signals like likes or saves, play duration offers a more reliable and verifiable measure of user interest. While some approaches construct weighted combinations of multiple engagement signals, we deliberately opt for a single, interpretable metric to avoid introducing volatility and platform-specific heuristics. The ground-truth distribution is defined as the aggregated play duration across future windows (3, 7, or 14 days). Although we cannot release raw correlation statistics due to data sensitivity and privacy constraints, internal audits confirm strong alignment between play duration and user feedback.

Our analysis targets high-activity users ($\geq$21 active days in the past month), as they represent the platform's core, most engaged cohort. Since we only have 30-days of user data, we pick high-activity users for their dense acitivities.

Durations are normalized per user to form a probability distribution over clusters, preventing heavy-usage magnitude from confounding preference shares.

## F ADDITIONAL EXPERIMENTAL RESULTS

### F.1 EVALUATION OF PET ON MOVIELENS

**Inference procedure of *Direct Generation*.** Unlike PET, which infers a full probability distribution over all 19 clusters, *Direct Generation* is prompted separately for each top-k values.

| Prediction | Method | Qwen3-8B | | | | DeepSeek-distill-Llama-8B | | | |
|---|---|---|---|---|---|---|---|---|---|
| | | Entropy@1 | Entropy@3 | Entropy@5 | Entropy@10 | Entropy@1 | Entropy@3 | Entropy@5 | Entropy@10 |
| Long | A: Direct Generation | 1.020 | 2.446 | 3.558 | 3.849 | 0.140 | 1.629 | 2.666 | 3.447 |
| | B: Logit-Probing | **1.747** | **3.130** | **3.565** | **3.979** | **2.307** | **3.054** | **3.352** | **3.813** |
| | C: Gen. Class. | 1.575 | 2.732 | 3.103 | 3.705 | 0.952 | 1.926 | 2.608 | 3.329 |
| Short | A: Direct Generation | .810 | 2.163 | 3.581 | 3.767 | 0.050 | 1.635 | 2.713 | 3.476 |
| | B: Logit-Probing | **2.093** | **3.246** | **3.612** | **3.964** | **2.256** | **2.922** | **3.252** | **3.701** |
| | C: Gen. Class. | 1.605 | 2.696 | 3.211 | 3.681 | 0.073 | 1.972 | 2.350 | 3.433 |

Table 3: Fairness performance on Movielens using the PT+SFT pipeline. We compare our Logit-Probing method (B) Generative Classification variant (C) against a Direct Generation benchmark (A). Each cell shows results for **Long-Term (L) / Short-Term (S)** preference. Metric: entropy score; context window: 3. Best results for each setting are in **bold**.

| Context | Prediction | Method | Qwen3-8B | | | | DeepSeek-distill-Llama-8B | | | |
|---|---|---|---|---|---|---|---|---|---|
| | | | NDCG@1 | NDCG@5 | NDCG@10 | JS Div | NDCG@1 | NDCG@5 | NDCG@10 | JS Div |
| 1 | Long | A: Direct Generation | **.813** | .573 | .644 | — | .609 | .440 | .540 | — |
| | | B: Logit-Probing | .781 | **.776** | **.823** | **.320** | **.734** | **.786** | **.840** | **.373** |
| | | C: Gen. Class. | .795 | .725 | .736 | .547 | .562 | .606 | .632 | .462 |
| | | Qwen3-Reranker-8B | .814 | .713 | .728 | .399 | | | | |
| | Short | A: Direct Generation | **.701** | .506 | .571 | — | .499 | .369 | .450 | — |
| | | B: Logit-Probing | .667 | **.675** | **.740** | **.466** | **.542** | **.619** | **.714** | **.514** |
| | | C: Gen. Class. | .638 | .661 | .789 | .538 | .499 | .490 | .501 | .551 |
| | | Qwen3-Reranker-8B | .650 | .566 | .652 | .525 | | | | |
| 3 | Long | A: Direct Generation | .820 | .573 | .644 | — | .601 | .428 | .533 | — |
| | | B: Logit-Probing | .815 | **.815** | **.858** | **.309** | **.758** | **.812** | **.812** | **.370** |
| | | C: Gen. Class. | **.830** | .760 | .758 | .547 | .637 | .624 | .624 | .462 |
| | | Qwen3-Reranker-8B | .809 | .714 | .727 | .401 | | | | |
| | Short | A: Direct Generation | **.702** | .495 | .554 | — | .492 | .361 | .453 | — |
| | | B: Logit-Probing | .686 | **.674** | **.707** | **.460** | **.501** | **.612** | **.712** | .507 |
| | | C: Gen. Class. | .661 | .631 | .671 | .539 | .492 | .499 | .507 | .545 |
| | | Qwen3-Reranker-8B | .624 | .563 | .645 | .525 | | | | |
| 5 | Long | A: Direct Generation | .840 | .538 | .629 | — | .604 | .428 | .542 | — |
| | | B: Logit-Probing | .830 | **.815** | **.825** | **.315** | **.763** | **.817** | **.867** | **.367** |
| | | C: Gen. Class. | **.847** | .777 | .764 | .547 | .423 | .528 | .551 | .463 |
| | | Qwen3-Reranker-8B | .802 | .709 | .725 | .403 | | | | |
| | Short | A: Direct Generation | **.723** | .496 | .546 | — | .493 | .359 | .454 | — |
| | | B: Logit-Probing | .716 | **.720** | **.781** | **.460** | **.496** | **.606** | **.708** | .507 |
| | | C: Gen. Class. | .673 | .635 | .676 | .535 | .489 | .498 | .507 | .546 |
| | | Qwen3-Reranker-8B | .606 | .552 | .641 | .525 | | | | |
| 8 | Long | A: Direct Generation | **.848** | .530 | .613 | — | .605 | .425 | .546 | — |
| | | B: Logit-Probing | .831 | **.822** | **.863** | **.316** | **.774** | **.823** | **.872** | **.367** |
| | | C: Gen. Class. | .846 | .760 | .762 | .542 | .362 | .518 | .542 | .465 |
| | | Qwen3-Reranker-8B | .801 | .701 | .721 | .406 | | | | |
| | Short | A: Direct Generation | **.711** | .499 | .529 | — | **.492** | .359 | .456 | — |
| | | B: Logit-Probing | .710 | **.719** | **.780** | **.462** | .490 | **.609** | **.711** | **.507** |
| | | C: Gen. Class. | .657 | .632 | .673 | .530 | .486 | .497 | .509 | .547 |
| | | Qwen3-Reranker-8B | .609 | .522 | .638 | .528 | | | | |

Table 4: Comprehensive performance on MovieLens using the PT+SFT pipeline across a range of context windows. We compare our Logit-Probing method (B) Generative Classification variant (C) against a Direct Generation benchmark (A) and Qwen3-Reranker-8B. Each cell shows results for **Long-Term (L) / Short-Term (S)** preference. Metric: NDCG. Best results for each setting are in **bold**.

## F.2 EVALUATION OF PET ON YELP

| Context | Prediction | Method | Qwen3-8B | | | DeepSeek-distill-Llama-8B | | |
|---|---|---|---|---|---|---|---|---|
| | | | Precision@1 | Precision@5 | Precision@10 | Precision@1 | Precision@5 | Precision@10 |
| 1 | Long | A: Direct Generation | **.996** | .847 | **.779** | .979 | .857 | .803 |
| | | B: Logit-Probing | .993 | **.951** | .754 | **.993** | **.966** | **.919** |
| | | C: Gen. Class. | .992 | .890 | .736 | .948 | .821 | .760 |
| | Short | A: Direct Generation | **.921** | .549 | .484 | .772 | .467 | .417 |
| | | B: Logit-Probing | .887 | **.711** | **.589** | **.788** | **.688** | **.608** |
| | | C: Gen. Class. | .882 | .627 | .517 | .772 | .588 | .434 |
| 3 | Long | A: Direct Generation | **.999** | .857 | .773 | .972 | .845 | .799 |
| | | B: Logit-Probing | .991 | **.960** | **.914** | **.996** | **.973** | **.926** |
| | | C: Gen. Class. | .997 | .911 | .792 | .897 | .854 | .755 |
| | Short | A: Direct Generation | **.926** | .555 | .488 | .775 | .466 | .429 |
| | | B: Logit-Probing | .909 | **.738** | **.617** | .755 | **.693** | **.618** |
| | | C: Gen. Class. | .902 | .654 | .518 | **.775** | .601 | .446 |
| 5 | Long | A: Direct Generation | .998 | .866 | .767 | .976 | .842 | .804 |
| | | B: Logit-Probing | .996 | **.961** | **.913** | **.993** | **.973** | **.931** |
| | | C: Gen. Class. | **.998** | .906 | .789 | .869 | .857 | .756 |
| | Short | A: Direct Generation | **.930** | .559 | .488 | **.768** | .463 | .429 |
| | | B: Logit-Probing | .926 | **.749** | **.622** | .746 | **.687** | **.620** |
| | | C: Gen. Class. | .912 | .662 | .525 | .762 | .597 | .443 |
| 8 | Long | A: Direct Generation | **.999** | .869 | .776 | .975 | .837 | .807 |
| | | B: Logit-Probing | .996 | **.963** | **.918** | **.998** | **.977** | **.934** |
| | | C: Gen. Class. | **.999** | .907 | .794 | .858 | .861 | .757 |
| | Short | A: Direct Generation | **.925** | .555 | .486 | **.774** | .461 | .431 |
| | | B: Logit-Probing | .922 | **.749** | **.621** | .736 | **.694** | **.622** |
| | | C: Gen. Class. | .907 | .658 | .521 | .762 | .597 | .447 |

Table 5: Comprehensive performance on Yelp using the PT+SFT pipeline across a range of context windows. We compare our Logit-Probing method (B) Generative Classification variant (C) against a Direct Generation benchmark (A). Each cell shows results for **Long-Term (L) / Short-Term (S)** preference. Metric: Precision. Best results for each setting are in **bold**.

| Context | Prediction | Method | Qwen3-8B | | | DeepSeek-distill-Llama-8B | | |
|---|---|---|---|---|---|---|---|---|
| | | | Recall@1 | Recall@5 | Recall@10 | Recall@1 | Recall@5 | Recall@10 |
| 1 | Long | A: Direct Generation | .072 | .299 | .547 | .071 | .301 | .563 |
| | | B: Logit-Probing | .072 | **.342** | **.637** | **.073** | **.350** | **.658** |
| | | C: Gen. Class. | .072 | .317 | .552 | .068 | .289 | .533 |
| | Short | A: Direct Generation | **.124** | .349 | .603 | .097 | .285 | .519 |
| | | B: Logit-Probing | .120 | **.461** | **.743** | **.106** | **.446** | **.772** |
| | | C: Gen. Class. | .118 | .402 | .654 | .097 | .367 | .538 |
| 3 | Long | A: Direct Generation | **.073** | .304 | .544 | .070 | .297 | .562 |
| | | B: Logit-Probing | .072 | **.345** | **.651** | **.073** | **.354** | **.666** |
| | | C: Gen. Class. | .072 | .325 | .559 | .064 | .302 | .531 |
| | Short | A: Direct Generation | **.124** | .354 | .608 | .096 | .278 | .527 |
| | | B: Logit-Probing | .122 | **.477** | **.777** | **.101** | **.443** | **.773** |
| | | C: Gen. Class. | .121 | .417 | .653 | .096 | .369 | .543 |
| 5 | Long | A: Direct Generation | .073 | .310 | .544 | .070 | .295 | .563 |
| | | B: Logit-Probing | .073 | **.349** | **.656** | **.072** | **.351** | **.666** |
| | | C: Gen. Class. | **.073** | .326 | .562 | .061 | .302 | .529 |
| | Short | A: Direct Generation | **.125** | .357 | .609 | .095 | .278 | .529 |
| | | B: Logit-Probing | **.125** | **.486** | **.784** | **.099** | **.441** | **.780** |
| | | C: Gen. Class. | .123 | .423 | .662 | .095 | .368 | .542 |
| 8 | Long | A: Direct Generation | **.073** | .309 | .547 | .070 | .292 | .564 |
| | | B: Logit-Probing | **.073** | **.348** | **.655** | **.073** | **.353** | **.668** |
| | | C: Gen. Class. | **.073** | .324 | .562 | .060 | .304 | .529 |
| | Short | A: Direct Generation | .125 | .356 | .607 | .096 | .277 | .532 |
| | | B: Logit-Probing | **.126** | **.488** | **.785** | **.099** | **.445** | **.781** |
| | | C: Gen. Class. | .123 | .422 | .659 | .095 | .370 | .547 |

Table 6: Comprehensive performance on MovieLens using the PT+SFT pipeline across a range of context windows. We compare our Logit-Probing method (B) Generative Classification variant (C) against a Direct Generation benchmark (A). Each cell shows results for **Long-Term (L) / Short-Term (S)** preference. Metric: Recall. Best results for each setting are in **bold**.

| Context | Prediction | Method | Qwen3-8B | | | | DeepSeek-distill-Llama-8B | | | |
|---|---|---|---|---|---|---|---|---|---|---|
| | | | NDCG@1 | NDCG@5 | NDCG@10 | JS Div | NDCG@1 | NDCG@5 | NDCG@10 | JS Div |
| 1 | Long | A: Direct Generation | 0.0015 | 0.1069 | 0.2430 | — | 0.0127 | 0.3109 | 0.3425 | — |
| | | B: Logit-Probing | **0.8604** | **0.8436** | **0.8669** | **0.6145** | **0.9581** | **0.8501** | **0.8777** | **0.6499** |
| | | C: Gen. Class. | 0.0945 | 0.1147 | 0.1587 | 0.6969 | 0.0000 | 0.1021 | 0.1437 | 0.7245 |
| | Short | A: Direct Generation | 0.0149 | 0.1133 | 0.2686 | — | 0.0127 | 0.2971 | 0.3333 | — |
| | | B: Logit-Probing | **0.7430** | **0.7467** | **0.7891** | **0.6768** | **0.7072** | **0.7352** | **0.7650** | **0.6969** |
| | | C: Gen. Class. | 0.0398 | 0.1122 | 0.1549 | 0.7301 | 0.0001 | 0.0928 | 0.1468 | 0.7296 |
| 3 | Long | A: Direct Generation | 0.0040 | 0.1164 | 0.3381 | — | 0.0170 | 0.3142 | 0.3450 | — |
| | | B: Logit-Probing | **0.9557** | **0.8724** | **0.9016** | **0.5976** | **0.9590** | **0.8683** | **0.8866** | **0.6527** |
| | | C: Gen. Class. | 0.1254 | 0.1230 | 0.1684 | 0.6935 | 0.0000 | 0.0647 | 0.1451 | 0.7179 |
| | Short | A: Direct Generation | 0.0277 | 0.1403 | 0.3556 | — | 0.0219 | 0.3034 | 0.3379 | — |
| | | B: Logit-Probing | **0.8743** | **0.7970** | **0.8548** | **0.6611** | **0.3391** | **0.4507** | **0.5194** | **0.7020** |
| | | C: Gen. Class. | 0.0454 | 0.1120 | 0.1577 | 0.7235 | 0.0028 | 0.0552 | 0.1823 | 0.7334 |
| 5 | Long | A: Direct Generation | 0.0115 | 0.1279 | 0.3522 | — | 0.0176 | 0.3136 | 0.3481 | — |
| | | B: Logit-Probing | **0.9629** | **0.8713** | **0.9075** | **0.5848** | **0.7814** | **0.8057** | **0.8350** | **0.6579** |
| | | C: Gen. Class. | 0.1204 | 0.1243 | 0.1720 | 0.6839 | 0.0004 | 0.0567 | 0.1554 | 0.7143 |
| | Short | A: Direct Generation | 0.1101 | 0.1472 | 0.3557 | — | 0.0203 | 0.3001 | 0.3394 | — |
| | | B: Logit-Probing | **0.8823** | **0.7923** | **0.8585** | **0.6511** | **0.6402** | **0.7237** | **0.7643** | **0.6944** |
| | | C: Gen. Class. | 0.0411 | 0.1137 | 0.1556 | 0.7194 | 0.0905 | 0.0802 | 0.1107 | 0.7317 |
| 8 | Long | A: Direct Generation | 0.0081 | 0.1390 | 0.3297 | — | 0.0156 | 0.3084 | 0.3429 | — |
| | | B: Logit-Probing | **0.9795** | **0.8787** | **0.9152** | **0.5734** | **0.9246** | **0.8563** | **0.8796** | **0.6546** |
| | | C: Gen. Class. | 0.1070 | 0.1183 | 0.1628 | 0.7012 | 0.0013 | 0.0615 | 0.1187 | 0.7168 |
| | Short | A: Direct Generation | 0.0739 | 0.1347 | 0.3152 | — | 0.0176 | 0.2948 | 0.3367 | — |
| | | B: Logit-Probing | **0.8593** | **0.7874** | **0.8599** | **0.6470** | **0.6809** | **0.7250** | **0.7637** | **0.6984** |
| | | C: Gen. Class. | 0.0402 | 0.1227 | 0.1526 | 0.7341 | 0.1095 | 0.0863 | 0.1292 | 0.7328 |

Table 7: Comprehensive performance on Yelp using the PT+SFT pipeline across a range of context windows. We compare our Logit-Probing method (B) Generative Classification variant (C) against a Direct Generation benchmark (A). Each cell shows results for **Long-Term (L) / Short-Term (S)** preference. Metric: NDCG. Best results for each setting are in **bold**. Level: L1

| Context | Prediction | Method | Qwen3-8B | | | DeepSeek-distill-Llama-8B | | |
|---|---|---|---|---|---|---|---|---|
| | | | Precision@1 | Precision@5 | Precision@10 | Precision@1 | Precision@5 | Precision@10 |
| 1 | Long | A: Direct Generation | 0.0054 | 0.4100 | 0.2583 | 0.1423 | 0.2503 | 0.2667 |
| | | B: Logit-Probing | **0.9538** | **0.6749** | **0.4897** | **0.9892** | **0.5025** | **0.3898** |
| | | C: Gen. Class. | 0.6152 | 0.3920 | 0.3456 | 0.0018 | 0.3508 | 0.2656 |
| | Short | A: Direct Generation | 0.0408 | 0.2252 | 0.1707 | 0.0444 | 0.2130 | 0.1885 |
| | | B: Logit-Probing | **0.8613** | **0.3718** | **0.2754** | **0.7905** | **0.3910** | **0.2628** |
| | | C: Gen. Class. | 0.1453 | 0.2006 | 0.1947 | 0.0006 | 0.1976 | 0.1681 |
| 3 | Long | A: Direct Generation | 0.0186 | 0.3880 | 0.2929 | 0.1531 | 0.2516 | 0.2672 |
| | | B: Logit-Probing | **0.9886** | **0.6379** | **0.4957** | **0.9946** | **0.5557** | **0.4017** |
| | | C: Gen. Class. | 0.7713 | 0.3863 | 0.3482 | 0.0006 | 0.2281 | 0.2606 |
| | Short | A: Direct Generation | 0.0828 | 0.2101 | 0.2060 | 0.0576 | 0.2172 | 0.1902 |
| | | B: Logit-Probing | **0.9526** | **0.3905** | **0.3186** | **0.4106** | **0.2403** | **0.2018** |
| | | C: Gen. Class. | 0.1477 | 0.1963 | 0.1983 | 0144 | 0.1005 | 0.1634 |
| 5 | Long | A: Direct Generation | 0.0690 | 0.3539 | 0.2918 | 0.1609 | 0.2549 | 0.2881 |
| | | B: Logit-Probing | **0.9898** | **0.6293** | **0.5030** | **0.9040** | **0.5651** | **0.4384** |
| | | C: Gen. Class. | 0.7863 | 0.4104 | 0.3592 | 0.0018 | 0.1878 | 0.1997 |
| | Short | A: Direct Generation | 0.3854 | 0.1894 | 0.2015 | 0.0540 | 0.2172 | 0.1975 |
| | | B: Logit-Probing | **0.9616** | **0.3864** | **0.3268** | **0.7419** | **0.4715** | **0.3124** |
| | | C: Gen. Class. | 0.1339 | 0.2110 | 0.1962 | 0.3025 | 0.1028 | 0.0850 |
| 8 | Long | A: Direct Generation | 0.0755 | 0.3220 | 0.3002 | 0.1519 | 0.2533 | 0.2757 |
| | | B: Logit-Probing | **0.9970** | **0.6293** | **0.5070** | **0.9676** | **0.5442** | **0.4164** |
| | | C: Gen. Class. | 0.6783 | 0.3903 | 0.3338 | 0.0078 | 0.2077 | 0.2478 |
| | Short | A: Direct Generation | 0.2636 | 0.1887 | 0.2048 | 0.0462 | 0.2108 | 0.1954 |
| | | B: Logit-Probing | **0.9652** | **0.3844** | **0.3413** | **0.8025** | **0.4415** | **0.2920** |
| | | C: Gen. Class. | 0.1429 | 0.2394 | 0.1838 | 0.3938 | 0.1122 | 0.1185 |

Table 8: Comprehensive performance on Yelp using the PT+SFT pipeline across a range of context windows. We compare our Logit-Probing method (B) Generative Classification variant (C) against a Direct Generation benchmark (A). Each cell shows results for **Long-Term (L) / Short-Term (S)** preference. Metric: Precision. Best results for each setting are in **bold**. Level: L1

| Context | Prediction | Method | Qwen3-8B | | | DeepSeek-distill-Llama-8B | | |
|---|---|---|---|---|---|---|---|---|
| | | | Recall@1 | Recall@5 | Recall@10 | Recall@1 | Recall@5 | Recall@10 |
| 1 | Long | A: Direct Generation | 0.0007 | 0.2977 | 0.3807 | 0.0173 | 0.1994 | 0.4070 |
| | | B: Logit-Probing | **0.1614** | **0.5224** | **0.7448** | **0.1675** | **0.4038** | **0.6063** |
| | | C: Gen. Class. | 0.0881 | 0.2823 | 0.5051 | 0.0002 | 0.2653 | 0.3910 |
| | Short | A: Direct Generation | 0.0089 | 0.2678 | 0.4435 | 0.0099 | 0.3070 | 0.4955 |
| | | B: Logit-Probing | **0.2604** | **0.5066** | **0.7148** | **0.2319** | **0.5153** | **0.6654** |
| | | C: Gen. Class. | 0.0304 | 0.2305 | 0.4533 | 0.0002 | 0.2313 | 0.3975 |
| 3 | Long | A: Direct Generation | 0.0025 | 0.2850 | 0.4488 | 0.0190 | 0.2032 | 0.4138 |
| | | B: Logit-Probing | **0.1699** | **0.4997** | **0.7660** | **0.1706** | **0.4500** | **0.6296** |
| | | C: Gen. Class. | 0.1149 | 0.2773 | 0.5120 | 0.0001 | 0.1672 | 0.3889 |
| | Short | A: Direct Generation | 0.0180 | 0.2496 | 0.5500 | 0.0117 | 0.3146 | 0.5038 |
| | | B: Logit-Probing | **0.2899** | **0.5252** | **0.8171** | **0.1288** | **0.3325** | **0.5308** |
| | | C: Gen. Class. | 0.0304 | 0.2214 | 0.4585 | 0.0030 | 0.1171 | 0.4046 |
| 5 | Long | A: Direct Generation | 0.0096 | 0.2624 | 0.4490 | 0.0196 | 0.2012 | 0.4378 |
| | | B: Logit-Probing | **0.1666** | **0.4903** | **0.7741** | **0.1532** | **0.4516** | **0.6763** |
| | | C: Gen. Class. | 0.1181 | 0.2950 | 0.5330 | 0.0002 | 0.1340 | 0.2853 |
| | Short | A: Direct Generation | 0.0926 | 0.2283 | 0.5344 | 0.0113 | 0.3070 | 0.5138 |
| | | B: Logit-Probing | **0.2859** | **0.5151** | **0.8345** | **0.2182** | **0.6206** | **0.7940** |
| | | C: Gen. Class. | 0.0272 | 0.2392 | 0.4568 | 0.0700 | 0.1174 | 0.1942 |
| 8 | Long | A: Direct Generation | 0.0105 | 0.2411 | 0.4689 | 0.0183 | 0.2012 | 0.4208 |
| | | B: Logit-Probing | **0.1729** | **0.4966** | **0.7887** | **0.1645** | **0.4377** | **0.6477** |
| | | C: Gen. Class. | 0.1016 | 0.2811 | 0.4921 | 0.0013 | 0.1484 | 0.3565 |
| | Short | A: Direct Generation | 0.0605 | 0.2223 | 0.5327 | 0.0097 | 0.3088 | 0.5182 |
| | | B: Logit-Probing | **0.2870** | **0.5137** | **0.8638** | **0.2392** | **0.5892** | **0.7530** |
| | | C: Gen. Class. | 0.0301 | 0.2784 | 0.4256 | 0.0908 | 0.1279 | 0.2726 |

Table 9: Comprehensive performance on Yelp using the PT+SFT pipeline across a range of context windows. We compare our Logit-Probing method (B) Generative Classification variant (C) against a Direct Generation benchmark (A). Each cell shows results for **Long-Term (L) / Short-Term (S)** preference. Metric: Recall. Best results for each setting are in **bold**. Level: L1

