# OpenReview forum: "PET: Preference Evolution Tracking with LLM-Generated Explainable Distribution"
_ICLR.cc/2026/Conference — ICLR 2026 Conference Withdrawn Submission_

### Official Review · Reviewer_Uu7r · 2025-10-28

**Soundness:** 1
**Presentation:** 2
**Contribution:** 2
**Rating:** 2
**Confidence:** 4

**Summary:**

This work proposes PET, a framework for user preference estimation via inferring a probability distribution over a fixed size of clusters that represent item classes. The proposed approach relies on unsupervised pre-training followed by supervised fine-tuning which approximates attractiveness scores for each of the clusters. During inference, the authors investigate likelihood-based probing compared to generative classification as well as hierarchical probing in case of very large cluster sets. Furthermore the authors provide a theoretical result based on optimality of the ranking via likelihood-probing based on certain assumptions. The experimental results showcase that likelihood-probing usually performs best, hierarchical probing performs well for large cluster sets and PET outperforms a sota embedding model for ranking. Finally, the authors provide improvements over a system used in production and show qualitative examples of evolving user preference clusters over time.

**Strengths:**

- The performance of PET seem strong compared to the baseline and the production system.
- The paper is easy to understand

**Weaknesses:**

**Conceptual framing**

The paper relies on the assumption that user preferences can be accurately captured merely by item categories and user history. I am not convinced that this information is sufficient. For example, recent work has shown the benefit of adding user reviews to approximate user preferences in a textual manner [1,2,3].  Furthermore, datasets like MovieLens actually provide additional information on user profiles that are neglected in its entirety in this work. Finally, the authors claim that PET constructs a user profile, but actually it is simply a sequence based classification based on clustering.

[1] Review-driven personalized preference reasoning with large language models for recommendation, Kim et al., SIGIR 2025

[2] Preference Discerning with LLM-Enhanced Generative Retrieval, Paischer et al., TMLR 2025

[3] USER-LLM: Efficient LLM Contextualization with User Embeddings, Ning et al., WWW 2025

**Theoretical results**

The theoretic optimality of PET ranking relies on a way too strong and unrealistic assumption, namely that the transformation from attractiveness score q to average logit score S is monotonic. While this assumption holds for the softmax function, there is an LLM in between that produces the logits for which such an assumption is not reasonable. This is corroborated by results showing that PET does not achieve the optimal ranking, which would be the case if this assumption was true.
I recommend to either (a) entirely remove the theoretical results as they are inconsistent and rely on a strong and unrealistic assumption, or (b), try to relax the assumption to an approximate or probabilistic monotonicity condition and re-derive the proof.

**Presentation**

The authors make plenty of claims around interpretability, unbiasedness, and fairness. However, the authors only show performance measures in terms of NDCG/Recall and no empirical evidence that supports improved interpretability/fairness. Especially claims on interpretability are heavily overclaimed as preference clusters merely consist of dataset-specific item catagories. The time evolution thereof is something that any other recommendation system can provide by visualizing item category distributions of predictions. Therefore I strongly recommend to caveat the claims on interpretability/fairness.

Another strong line of claims is that PET was designed for recommending niche/long-tail interests, but it is unclear to me what design choices in particular facilitate recommendation of long-tail interests? In particular, many other recommendation systems, especially generative retrieval based ones [1], support that by simply increasing the temperature during decoding, why is this special in PET?

[1] Recommender Systems with Generative Retrieval, Rajput et al., NeurIPS 2023

**Baselines**

The authors only compare to different decoding/prompting schemes of the same model and to one other model, but do not consider state-of-the-art generative recommendation systems in the literature. In this regard semantic-ID based methods [1,2,3,4] would be particularly interesting as they promise to provide a similar hierarchical clustering of items as handcrafted for Table 1. Furthermore, other logit-based ranking methods already exist which could also be included in the comparison [5]. Finally, comparing to a production system (Table 2) without disclosing any details about its implementation does have minimal scientific value. Therefore I recommend including the suggested baselines.

[1] Recommender Systems with Generative Retrieval, Rajput et al., NeurIPS 2023

[2] Preference Discerning with LLM-Enhanced Generative Retrieval, Paischer et al., TMLR 2025

[3] Unifying Generative and Dense Retrieval for Sequential Recommendation, Yang et al., TMLR 2025

[4] Adapting Large Language Models by Integrating Collaborative Semantics for Recommendation, Zheng et al., ICDE 2024

[5] Generative Sequential Recommendation with GPTRec, Petrov et al., SIGIR 2023

**Experiment Design**

The authors claim that Table 2 evaluates for long-tail interests, but the experimental design of this scenario does not necessarily require modelling long-tail interests. In fact, this experiment is about long-term predictions (7-14 days), but there is no supporting analysis of this evaluation scenario that ground-truth items in that duration are actually long-tailed ones and not also popular ones. Supporting this claim requires some evidence, like showing the distribution of items of test splits compared to training splits.

**Minor Remarks**

- Table 1: Logit Probing should be replaced with Hierarchical probing, otherwise it makes the wrong impression that vanilla logit probing is used here.

**Questions:**

- What is meant by the task of user segmentation?
 - Line 127: What is meant by a "transparent distribution"?
 - Line 283: What is the average lifetime for a user for the different datasets? How much longer are those timelines compared to next-item prediction?
 - Line 318: Does the max of 8 sessions mean that the context window at most comprises the last 8 interactions of the user? If so, why was this threshold chosen? Wouldn't it be more beneficial to allow longer context windows especially for long-term predictions?
- Table 1: are the scores only based on predicting the correct L1/L2 cluster? or also for predicting the correct item in L2 cluster? if it is only on abstract cluster level, the item-level scores should be added, prediction on cluster level is a lot easier than predicting items within the cluster.

---

> ### Author Response · Authors · 2025-12-03
>
> **On Conceptual framing**
>
> We thank the reviewer for raising this point and for highlighting recent work. In our paper, a user profile is not a static textual biography but a **dynamic probability distribution over preference clusters**. PET is trained to match the empirical future distribution​ over a horizon, i.e., to perform density estimation over the full preference space (including long-tail intensity), rather than predicting a single next item as in standard sequence classification. This distributional state is then reused for ranking, temporal evolution, and diagnostics, instead of being implicitly encoded in a black-box embedding.
>
> We also wish to clarify that, our implementation **does incorporate** available reviews and profile fields into the input histories. As illustrated in Fig. 1, the history encoder takes as input not only item-level interaction sequences but also associated user reviews and user-side textual fields when they are provided by the dataset. In the revision we update the experimental design to explicitly state which review and profile attributes are included for training on MovieLens/Yelp dataset.
>
> Conceptually, we view the cited works [1–3] as **complementary but operating at a different level of the stack**. Kim et al. and Paischer et al. focus on contextualization and retrieval for end-to-end recommendation (improving the **input representation and candidate generation/ranking objective**), and USER-LLM learns user embeddings for efficient LLM conditioning. By contrast, PET targets the **user modeling layer**: given whatever contextualization the system chooses (including reviews, embeddings, or retrieved documents), PET’s contribution is to produce an explicit, interpretable output distribution over preference clusters. This distribution can then be consumed by downstream recsys components (including the architectures in [1–3]) for candidate generation, re-ranking, or auditing. We clarify this positioning in the revised manuscript so that PET is understood as a distributional user-modeling framework that can plug into, rather than replace, existing LLM-based recommendation pipelines. And we also cite your mentioned papers in our related work part.

---

> ### Author Response · Authors · 2025-12-03
>
> **On Theoretical Results**
>
> We appreciate the reviewer’s critique regarding the isotonicity assumption. This feedback motivated us to formalize the theoretical underpinnings of PET. In the revision, we demonstrate that PET is grounded in statistical learning theory, providing guarantees that hold even when the idealized assumption of perfect calibration is relaxed. We have expanded Section 3.3 and Appendix D into a rigorous two-layer framework that directly addresses your concerns:
>
> 1. Isotonicity is Derived, Not Assumed (Fisher Consistency)
>
>    - We clarify that the monotonicity between logits and preferences is not a heuristic assumption but the theoretical limit of the training objective.
>
>    - By leveraging the theory of Strictly Proper Scoring Rules (Gneiting & Raftery, 2007), we prove that in the population limit, minimizing cross-entropy guarantees the recovery of the true conditional preference distribution (Proposition 1).
>
>    - At this global optimum, strictly sorting logits is equivalent to sorting latent utilities. We present this as an idealized Bayes-optimal limit (Fisher Consistency), establishing the mathematical validity of the objective without claiming finite models achieve it.
>
> 2. Adopting Probabilistic Relaxation
>
>    - To address the "imperfect" regime, we explicitly followed your recommendation to "relax the assumption to an approximate or probabilistic monotonicity condition." We introduce the $\epsilon$-approximate pairwise isotonicity condition (Assumption 1).
>
>    - Under this relaxed assumption, we derive a formal bound on the Pairwise Ranking Regret of $O(K^2 \epsilon)$ (Theorem 1). This confirms that the ranking quality degrades gracefully with model error, rather than relying on an "all-or-nothing" assumption.
>
> 3. Connecting Theory to Learning Dynamics
>    - We further defend the methodology by linking the training loss to the ranking guarantee.
> Using Pinsker’s Inequality (Lemma 1), we prove that minimizing the excess cross-entropy risk minimizes the upper bound on the ranking regret.
>    - This provides a principled interpretation: \epsilon is not a fixed property of the LLM, but a residual error driven down by training. Lower training loss mathematically guarantees a tighter bound on ranking regret.
>
> 4. Reframing the Contribution
>    - We have updated the main text to reflect this rigorous scope. The theory is now presented as (i) a clear Bayes-consistent limit, and (ii) a controlled error bound in the approximate regime. We believe this successfully replaces the original heuristic argument with the rigorous probabilistic framework you requested.

---

> ### Author Response · Authors · 2025-12-03
>
> **On interpretability, fairness, and long-tail design.**
>
> We respectfully disagree with the reviewer’s implication that our claims about interpretability and fairer personalization are unsupported. PET is explicitly formulated as a distributional user model: its primary output is a probability vector over human-readable clusters, trained to match the empirical future distribution of user interactions, not just a top-k item list. This means that the user’s preference state is represented as an explicit probability distribution that can be directly inspected, segmented, and tracked over time, independently of any particular retrieval or ranking stack. By contrast, “visualizing category distributions” from arbitrary recommender outputs operates on the exposure induced by a specific system; it does not give access to the model’s internal belief over clusters and conflates interest with candidate generation. PET’s design is precisely to make that latent preference distribution a first-class prediction target.
>
> In response to this and a related concern from Reviewer zU4r about “qualitative” fairness, we have added quantitative diversity and calibration analyses to the revised manuscript. First, we compute Global Exposure Entropy by aggregating top-k ranked clusters across users and measuring the entropy of the resulting exposure distribution. PET consistently achieves higher entropy than Direct Generation (DG) while simultaneously attaining higher NDCG/Recall, demonstrating that it reduces concentration on a few head clusters without trading off accuracy. Second, we evaluate calibration in the sense standard in recommendation (distributional alignment between predicted preferences and realized behavior) via the Jensen–Shannon Divergence between each method’s predicted preference distribution and the empirical next-window distribution. PET attains the lowest JS-Div on all datasets, indicating that its inferred probability mass is more faithfully aligned with users’ actual future preferences than DG or the reranker baseline. For small public cluster spaces (e.g., 19 MovieLens genres), naive coverage@k quickly saturates for all methods and becomes non-discriminative; this is why we focus on entropy and JS-Div rather than raw coverage. These metrics directly contradict the notion that there is “no empirical evidence” for less head-dominated, better-aligned personalization.
>
> Regarding niche/long-tail interests and the suggestion that simply increasing temperature in a generative-retrieval system [4] achieves the same effect, we again disagree. Temperature is a coarse decoding knob: it operates only at sampling time, flattens probabilities indiscriminately, and typically increases entropy by injecting noise, without producing a reusable representation of user preferences. PET’s behavior is driven by structural choices in both training and inference. (i) PET is trained on a cluster-level distribution aggregated over a future horizon, so sustained engagement with niche clusters is explicitly accumulated into the target distribution even when individual items are rare. (ii) At inference, PET uses pre-softmax logits with deterministic ranking, avoiding beam-search truncation and sampling effects that suppress low-probability clusters in autoregressive decoding; any cluster that receives non-negligible mass will be reflected in the ranked distribution. These properties are orthogonal to, and in fact complementary with, generative retrieval: PET can supply a calibrated, interpretable preference distribution that guides or constrains GR-based candidate generation (including any temperature settings), rather than being reducible to a single decoding hyperparameter. We will make these design choices and their implications for long-tail recommendation explicit in Section 4.4 of the revised manuscript.

---

> ### Author Response · Authors · 2025-12-03
>
> **On Experiment Design and Validation of Long-Tail Capture**
>
> We thank the reviewer for raising this point. The reviewer hypothesizes that our long-term evaluation (Table 2) might be dominated by popular items simply because the time window is longer, and asks for evidence that the ground truth is actually long-tail. However, we respectively clarify that this critique stems from a misunderstanding of our evaluation protocol.
>
> - **Explicit Filtering, Not Just Time Windows.** The results in Table 2 are not generic NDCG scores over the full item space. We explicitly defined a "Long-Tail Segment" (the bottom 25% of clusters by global interaction volume) and calculated the reported metrics only on the subset of test instances where the ground-truth interaction belonged to this segment. By definition, every positive sample in this evaluation is a niche interest. It is mathematically impossible for "popular ones" to contaminate this metric, as they are excluded from the ground-truth set for this specific analysis.
>
> - **Why Long-Term Matters for the Tail.** We evaluated this specifically over a long-term horizon (7-14 days) because long-tail interests (e.g., specific hobbies like "Woodworking" or "Indie Horror") represent stable, intrinsic user traits. In contrast, short-term history is often dominated by algorithmic noise and viral trends (global popularity). By predicting the aggregate distribution over a 14-day future window, PET filters out short-term viral noise and recovers the user’s persistent niche interests. The high NDCG scores in Table 2 confirm that PET successfully identifies these niche clusters when they occur, whereas the production model—optimized for short-term viral clicks—fails to surface them.
>
> **Minor remarks**
>
> We thank the reviewer for the suggestion. To clarify, all results in Table 1 utilize the Hierarchical Probing framework (Algorithm 3) to manage the high-dimensional Yelp cluster space. The labels distinguish the scoring mechanism applied at each node of the hierarchy: the row currently labeled "Logit Probing" refers to the Hierarchical framework using Likelihood-based Probing (Algorithm 1) as the node scorer. We will update the labels in the revision to "Hierarchical (Logit)" and "Hierarchical (Generative)" to explicitly reflect this composition and avoid confusion with flat probing.

---

> ### Author Response · Authors · 2025-12-03
>
> **Questions**
>
> Q: What is meant by the task of user segmentation?
>
> A: User segmentation refers to grouping users based on the similarity of their inferred preference distributions. Because PET represents users as explicit probability vectors over interpretable clusters (e.g., [0.8 Horror, 0.2 Comedy]), operators can cluster these vectors to identify coherent user tribes (e.g., "Niche Horror Enthusiasts") for targeted analysis. This is difficult with opaque latent embeddings.
>
> Q: Line 127: What is meant by a "transparent distribution"?
>
> A: "Transparent" means the model outputs explicit probabilities over human-readable labels (e.g., "Sci-Fi: 60%, Drama: 40%") rather than a high-dimensional latent vector or a black-box logits score for an arbitrary Item ID. This transparency allows for direct inspection, fairness auditing (e.g., checking if "News" is suppressed), and debugging without requiring auxiliary probe networks.
>
> Q: Line 283: User lifetime vs. next-item prediction?
>
> A:
> - **Lifetimes:** In MovieLens and Yelp, user timelines span years (averaging ~2-6 years in standard splits). In the TikTok dataset, the "long-term" evaluation covers a 14-day future window.
> - **Comparison:** "Next-item" prediction looks at a single interaction step $(t+1)$, often seconds or minutes into the future. Our long-term evaluation covers the aggregate distribution of all interactions over the remaining lifetime or window $(t+1 \dots T)$. The scale difference is orders of magnitude (single event vs. hundreds/thousands of events), designed to capture stable traits rather than transient clicks.
>
> Q: Line 318: Why max 8 sessions? Wouldn't longer be better?
>
> A: "8 Sessions" refers to interaction sessions, each containing multiple items. A context window of 8 sessions typically encompasses dozens to hundreds of specific interactions, filling the effective context length of the LLM. We capped this threshold because:
> - **Efficiency:** Inference costs scale with context length.
> - **Diminishing Returns:** Empirical studies on LLMs (e.g., "Lost in the Middle" phenomena) and sequential recommendation suggest that very old history often adds noise or is less predictive of immediate future interests than recent sessions.
>
> Q: Table 1: Are the scores only based on predicting the correct L1/L2 cluster? ... prediction on cluster level is a lot easier than predicting items...
>
> A: **We respectfully but fundamentally disagree with the premise that cluster-level prediction is merely an "easier" version of item prediction.** This critique conflates User Modeling (our contribution) with Candidate Retrieval (a downstream task). We chose cluster-level evaluation for three specific methodological reasons:
>
> - **The "User Profile" Definition:** As defined in Section 3.1, the output of PET is a probability distribution over the semantic preference space, not a list of Item IDs. Evaluating PET on Item-ID hits (e.g., HR@20) would require attaching an arbitrary retrieval module (e.g., fetching items that match the predicted cluster). This would introduce a confounding variable: a low item score could result from a poor retriever, obscuring the quality of the PET user model. Table 1 isolates the quality of the preference distribution itself.
>
> - **Stability vs. Stochastic Noise:** The reviewer suggests item prediction is "harder," but in many cases, it is simply noisier. Predicting exactly which specific 30-second video or restaurant ID a user will interact with 7 days from now is dominated by aleatoric uncertainty (randomness) and platform serving logic. In contrast, predicting that a user allocates 60% of their interest to "Italian Food" and 40% to "Mexican Food" captures a stable, intrinsic user trait. A model that chases Item IDs often overfits to short-term history; PET aims to capture these stable long-term traits.
>
> - **Generalization vs. Memorization:** In high-dimensional spaces (like Yelp's 1,311 categories vs. millions of items), item-level prediction often relies on memorizing co-occurrence patterns (Collaborative Filtering). Cluster-level prediction requires the model to generalize -- understanding that a user who liked "Inception" (Item A) has an interest in "Sci-Fi" (Cluster A), and therefore might like "Interstellar" (Item B) even if they have never seen it. By evaluating at the cluster level, we validate that the LLM has learned the semantics of the user's preference, which is critical for long-tail discovery and cold-start recommendation, where item-level signals are sparse or non-existent.
>
> **References**
>
> [1] Review-driven personalized preference reasoning with large language models for recommendation, Kim et al., SIGIR 2025
>
> [2] Preference Discerning with LLM-Enhanced Generative Retrieval, Paischer et al., TMLR 2025
>
> [3] USER-LLM: Efficient LLM Contextualization with User Embeddings, Ning et al., WWW 2025
>
> [4] Recommender Systems with Generative Retrieval, Rajput et al., NeurIPS 2023

---

### Official Review · Reviewer_PifA · 2025-10-31

**Soundness:** 3
**Presentation:** 3
**Contribution:** 3
**Rating:** 6
**Confidence:** 3

**Summary:**

This paper introduces Preference Evolution Tracking (PET), a new framework that reframes LLM-based personalization by moving beyond direct item ranking. By inferring a dynamic probability distribution over a stable lattice of interpretable preference clusters, PET is designed to achieve a more holistic understanding of user interests. This distributional profile is inferred using scalable techniques like Likelihood-based Probing and Generative Classification, while Hierarchical Probing ensures the framework can handle thousands of clusters in real-world applications. A crucial outcome of the model is a transparent and portable user profile, which serves as an auditable asset that mitigates popularity bias by effectively capturing and ranking users' niche, long-tail preferences. Comprehensive evaluation on public benchmarks like MovieLens and Yelp demonstrates NDCG improvements of up to 40% over direct generation, and validation on an industrial-scale dataset reveals a 7-fold improvement in ranking long-tail content over a state-of-the-art production model. Further analysis confirms PET's ability to generate interpretable, evolving user profiles, adeptly capturing the dynamics of both stable, long-term interests and volatile, context-dependent preferences over time.

**Strengths:**

1. The paper is well-structured with clear writing and is supported by rich and lucid illustrations that effectively communicate the framework and results.

2. The proposed paradigm shift from direct item ranking to distributional preference inference is a significant conceptual contribution that thoughtfully addresses well-known and critical challenges in modern recommender systems, such as popularity bias and lack of transparency.

3. The framework's practical value is substantiated by exceptional empirical results, particularly the substantial performance improvement on long-tail content in a large-scale industrial setting, which provides compelling evidence of its real-world impact beyond academic benchmarks.

4. The methodology is practical and scalable, offering a thoughtful suite of inference strategies (Likelihood-based Probing, Generative Classification) and a pragmatic solution for high-dimensional spaces (Hierarchical Probing) that shows clear consideration for deployment challenges.

**Weaknesses:**

1. **Ambiguity in the Mechanism for Countering Popularity Bias:** A central claim of the paper is that the PET framework effectively counters popularity bias and improves the discovery of long-tail user interests. However, there is a conceptual tension in the methodology that requires further clarification. The ground-truth "proxy preference distribution," as defined in Equation (3), is calculated from the normalized frequency of a user's historical
interactions within each cluster. Since interaction frequency is inherently a measure of popularity (albeit at a personal level), the model is being trained to replicate a signal that is itself influenced by popularity. The paper would be substantially strengthened if the authors explicitly elucidated the mechanism by which a model trained on this frequency-based target can successfully counter global popularity bias.

2. **Ambiguity in the Validation of Long-Tail Preference Capture:** While the results of proposed methods in Table 2 are impressive, the manuscript does not provide a clear and explicit definition of how the "long-tail segment" of TikTok dataset was constructed for the evaluation.  For the reader to fully assess the significance of this core claim, it is crucial to understand what constitutes the "long-tail" in this experiment

**Questions:**

1. **On the Rationale for Hierarchical Probing's Top-Down Approach:** Could the authors elaborate on the rationale for Hierarchical Probing's top-down (coarse-to-fine) strategy, as a bottom-up approach might seem more intuitive? What are its specific advantages for computational tractability and scalability in extreme multi-label classification scenarios?

2. **On the Mechanistic Difference Between Generative Classification and Direct Generation:** Given the similar prompts, could the authors provide a simple example to illustrate the output difference between Generative Classification and Direct Generation

---

> ### Author Response · Authors · 2025-12-03
>
> **On "Ambiguity in the Mechanism for Countering Popularity Bias"**
>
> We thank the reviewer for raising this point. We clarify (i) what signal PET is trained on in Eq. (3), (ii) how PET relates to existing work on de-biasing logged data, and (iii) which aspect of popularity bias PET is designed to mitigate.
>
> - **User-normalized preference vs global popularity.**
>   Eqn (3) essentially calculates the user level frequency, which is different from global popularity. Thus a cluster can be globally rare but still have high probability for a user if it makes up a large fraction of their history (e.g., 5 "viral" and 5 "niche" items ⇒ [ ⁣0.5,0.5 ⁣] for that user). PET is therefore modeling personal preference distributions, not global counts.
>
> - **Pet is complementary to existing de-bias literature.**
>   We agree that the underlying interaction counts are shaped by exposure bias, and a substantial literature addresses this by re-weighting or regularizing the training objective (e.g., IPS / unbiased implicit-feedback / popularity-aware ranking [1-3]). Our goal in this work is orthogonal: we do not propose a new debiasing scheme for the logs, but instead focus on how to avoid amplifying this bias at inference.
>
> - **Mitigating inference/decoding amplification of popularity.**
>   Concretely, Direct Generation uses autoregressive decoding, which is mode-seeking and tends to truncate the probability tail (niche clusters rarely “win” the beam). PET instead directly reads and normalizes logits for all clusters, preserving non-zero probability for long-tail interests and allowing them to be ranked. In TikTok dataset, where we define "long-tail" as the bottom 25% clusters by global viewing duration, PET achieves NDCG@20 = 0.197 vs. 0.024 for the production model on this tail subset, consistent with this mechanism.
>
> **On "Ambiguity in the Validation of Long-Tail Preference Capture"**
>
> We thank the reviewer for pointing out that our definition of the “long-tail segment” was not explicit. In the TikTok experiment, we operate over 78 content clusters defined by the platform’s internal interest taxonomy. We define long-tail clusters as the bottom 25% of clusters when ranked by global viewing duration; in our data this corresponds to 20 out of 78 clusters.
>
> For the long-tail evaluation in Table 2, we restrict to test instances where the ground-truth next interaction belongs to this long-tail cluster set, and compute NDCG@20 on this subset using the same prediction lists as in the main TikTok metrics. Thus, the reported numbers measure how well each method recovers long-tail clusters that actually occur in user behavior, rather than performance on the head.

---

> ### Author Response · Authors · 2025-12-03
>
> **On the Rationale for Hierarchical Probing's Top-Down Approach**
>
> We thank the reviewer for this question. We chose a top-down (coarse-to-fine) strategy in Hierarchical Probing for both computational and modeling reasons.
>
> From a computational perspective, a bottom-up or flat approach would require probing all fine-grained clusters at once, which scales linearly with the total number of labels and quickly becomes expensive in extreme multi-label settings (e.g., Yelp). Our top-down procedure first probes a small number of coarse L1 clusters and then only expands the few highest-probability branches. This effectively acts as a hard attention mechanism over the taxonomy: we filter out irrelevant domains (e.g., Automotive) when the user is clearly in Food, and only spend compute on fine-grained scoring within the relevant subtrees. In practice, this leads to far fewer probes than scoring all K clusters for every user.
>
> From a modeling perspective, coarse L1 categories are more semantically stable and easier for the LLM to reason about than hundreds of fine-grained labels in a single pass. Conditioning later probes on the selected L1 branch (e.g., “within Food & Restaurants, which sub-clusters apply?”) provides additional context and yields more calibrated, less noisy probabilities over L2/L3 clusters. The top-down design also gives operators explicit control over how deep to expand different branches (e.g., only refine certain domains when exploring diversity), which would be difficult with a flat or purely bottom-up scheme.
>
> **On the Mechanistic Difference Between Generative Classification and Direct Generation**
>
> We thank the reviewer for this question. Conceptually, Direct Generation treats recommendation as open-ended sequence generation: we prompt the LLM to “list” the user’s preferred clusters and use autoregressive decoding (e.g., greedy/beam search) to produce a short list. Generative Classification, in contrast, constrains the output to a fixed set of candidate clusters and uses the model’s log-likelihoods over these options as scores, i.e., it uses the LLM as a probabilistic classifier rather than as a free-form generator (similar in spirit to using LLMs for label scoring via verbalizers).
>
> As a simple example, suppose the model’s internal belief over three clusters is P(Action)=0.6,  P(Romance)=0.3,  P(Documentary)=0.1. With Direct Generation, a prompt such as “List this user’s favorite genres” and beam search will typically produce a sequence like “Action, Romance”; the 10%-probability “Documentary” cluster may never appear, and its signal is effectively lost. With Generative Classification, we instead score each candidate cluster (e.g., via an option list or yes/no prompt per cluster), read the logits, and normalize them to recover a distribution close to (0.6,0.3,0.1). PET operates in this latter regime: we always obtain a full probability distribution over clusters, so even lower-probability, long-tail interests like “Documentary” receive explicit non-zero scores and can be surfaced, rather than being discarded by argmax decoding.
>
> **References**
>
> [1] Schnabel et al., 2016, Recommendations as Treatments: Debiasing Learning and Evaluation, ICML.
>
> [2] Saito et al., 2020, Unbiased Recommender Learning from Missing-Not-At-Random Implicit Feedback, WSDM.
>
> [3] Abdollahpouri et al., 2017, Controlling Popularity Bias in Learning-to-Rank Recommendation, RecSys.

---

### Official Review · Reviewer_zU4r · 2025-11-01

**Soundness:** 2
**Presentation:** 2
**Contribution:** 2
**Rating:** 2
**Confidence:** 4

**Summary:**

This paper introduces a framework to model evolving user preference of LLMs in an interpretable manner. By applying logit probing and generative classification techniques, it models user preference as a probability distribution over interpretable clusters instead of generating ranked lists of items. Experiments are performed on MovieLens, Yelp, and a proprietary TikTok dataset to validate the method.

**Strengths:**

1. The idea of interpretable and distributional personalization with LLMs sounds to be interesting. Authors reframes LLM-based personalization from direct ranking to distributional inference. This conceptual shift allows interpretability.
2. Empirically, the performance gain in terms of NDCG is appealing.

**Weaknesses:**

Overall, the paper’s core value lies in its interpretability and long-tail performance, not in fundamental algorithmic contribution.

1. The core methods introduced in section 3, i.e. logit probing and generative classification, are directly borrowed from existing interpretability work (Petroni et al., 2019; Schick & Schütze, 2020). The main contribution is contextual repurposing to user modeling, and the technical novelty is limited in terms of methodology.

2. The claim of fairer personalization is qualitative. Evaluations on diversity/fairness, which are central to the paper’s motivation, are not quantitatively analyzed (no metrics such as coverage, entropy, or calibration).

3. Authors attempted to provide theoretical arguments for the proposed methodology, but the theoretical substance is limited.
- The proof in Appendix D is rather heuristics-based and not rigorous in any sense.
- The “optimality” theorem (Lemma 1) is straightforward and expected under the isotonic assumption, which essentially assumes what needs to be proved (i.e., correct ordering of logits).
- In terms of the theoretical contributions, it did not provide insights in terms of learning or generalization guarantee of the methodology.
As such, to me, section 3.3 is insufficiently grounded and would benefit from a more rigorous justification of its claims.

4. The computational overhead of Likelihood-based Probing appears large (O(K) forward passes), even with KV reuse. Author may want to include runtime comparisons to further demonstrate the practical applicability of the method.

5. While the TikTok dataset is proprietary and anonymized, current details on feature engineering, taxonomy creation, and production baselines are not clear from the manuscript. Authors may want to include more detailed info of the dataset.

I am willing to defend my decision if authors can sufficiently address the main weaknesses.

**Questions:**

1. PET assumes predefined cluster taxonomies. How sensitive is performance to the choice or granularity of these clusters? Could automatic or learned clustering be incorporated?

**Details Of Ethics Concerns:**

No ethics concerns. Ethics statement is provided.

---

> ### Author Response · Authors · 2025-12-03
>
> **On novelty and contribution**
>
> We thank the reviewer for raising this question and acknowledge that logit probing and generative classification are established primitives, and we do not present them as novel operators. In PET, we deliberately treat them as building blocks. Petroni et al. (2019) and Schick & Schütze (2020) are cited as influential applications of these ideas in interpretability and few-shot classification, not as the origin of using language-model logits/likelihoods to score labels -- a practice that predates these works and is now standard across many domains.
>
> However, we **respectfully disagree** that PET’s contribution is "only contextual repurposing." Our methodological novelty lies in how these primitives are instantiated and combined to solve a different problem:
>
> - **New problem formulation.** PET casts LLM-based personalization as inferring a full probability distribution over interpretable preference clusters from a user’s history, trained to match empirical future distributions. This distribution can be reused across horizons and tasks (ranking, evolution analysis, long-tail diagnostics), which differs from prior LLM+Rec work that focuses on next-item generation in the item-ID space.
>
> - **Hierarchical Probing for extreme multi-label spaces.** To handle extreme numbers of clusters (e.g., 1000+ Yelp categories), we introduce Hierarchical Probing: a two-stage, tree-structured inference scheme that first scopes L1 categories and then selectively probes L2 children. Together with the complexity analysis in Appendix C and the Yelp results in Section 4, this shows that PET remains tractable and stable in high-dimensional preference spaces where flat iterative probing becomes expensive.
>
> - **Refined theoretical and structural analysis.** In the revision, Section 3.3 and Appendix D formalize the statistical underpinnings of PET: (i) deriving isotonicity as a Fisher-consistent limit of cross-entropy training under a latent-softmax model, (ii) an $\epsilon$-approximate regime with an $O(K^2 \epsilon)$ pairwise-ranking regret bound linked to excess cross-entropy, and (iii) a structural comparison explaining why decoding-based direct generation is more susceptible to calibration/frequency bias, search truncation, and exposure bias.
> We have revised the manuscript to make this positioning explicit: PET builds on standard probing primitives, but the core contribution lies in its distributional user-modeling formulation, the hierarchical probing architecture for large label spaces, and a theoretically grounded structural analysis that explains PET’s empirical gains over direct generation.
>
> We have revised our manuscript to make this positioning explicit: PET builds on standard probing primitives, but the contribution lies in its distributional formulation of user modeling, the hierarchical probing architecture for large label spaces, and a refined theoretical/structural analysis that connects the training objective to ranking behavior and explains PET’s empirical gains over direct generation.

---

> ### Author Response · Authors · 2025-12-03
>
> **On Fairness Metrics**
>
> We thank the reviewer for suggesting several fairness metrics, and below is our analysis:
>
> **Coverage.** We agree that coverage is a standard diversity signal for item-level recommendation. However, in our cluster-level setting, we find that coverage is not a discriminative metric since PET could essentially cover the entire cluster space. For Direct Generation (DG), the coverage rate is also not an intrinsic property of the model: we can always force DG to output a wider set of clusters by increasing parameter such as temperature, top-$k$, top-$p$ during sampling, but our empirical results show this comes at a substantial cost in ranking quality (which we also observe empirically in our experiments in Section 4). Consequently, realistic deployments use DG only for moderate list lengths (e.g., $k \le 20$). In this regime, and given the small cluster spaces (e.g., 19 genres in MovieLens), both PET and DG expose almost all clusters across users, so cluster coverage@k quickly saturates and becomes non-informative. For these datasets, we therefore focus on order-aware metrics (NDCG/Recall/Precision) that directly capture ranking quality over the cluster space, and reserve more detailed diversity/long-tail analyses for TikTok, where the category lattice is large and heavily long-tailed.
>
> **Entropy.** To quantitatively assess distributional fairness, we added a Global Exposure Entropy metric. For each method and cutoff k, we aggregate the top-k ranked clusters over all users, form the resulting exposure distribution over the cluster space, and compute its Shannon entropy (higher values mean exposure is spread more evenly across clusters rather than concentrated on a few popular ones). Table 1 shows that PET consistently achieves higher exposure entropy than DG across both backbones and both short- and long-term settings.
>
> - **At k = 1.** On DeepSeek-Distilled-Llama-8B, DG exhibits severe mode collapse, with Entropy@1 as low as 0.140 bits, indicating that the top-1 recommendation is dominated by essentially the same 1–2 popular genres for almost all users. In contrast, our Logit-Probing PET variant achieves Entropy@1 of 2.307, a more than 16× increase in entropy, corresponding to a much larger effective number of exposed clusters. An analogous gap appears under Qwen3-8B (1.020 vs. 1.747). These results show that PET surfaces more user-specific and diverse primary recommendations instead of collapsing onto a small set of head clusters.
>
> - **At k = 10.** As expected, given the finite cluster space (19 MovieLens genres), Entropy@10 begins to saturate for all models. However, at this saturation point, PET still maintains higher entropy (for example, 3.979 vs. 3.849 for Qwen3-8B long-term, and 3.813 vs. 3.447 for DeepSeek) and significantly better ranking accuracy (NDCG/Recall). This indicates that PET achieves greater exposure diversity through better-aligned personalization, rather than by injecting random noise into the recommendation lists.
> Overall, the Global Exposure Entropy results quantitatively support our claim that PET provides fairer, less head-dominated personalization, complementing the accuracy gains reported in Section 4.
>
> **Calibration.** We thank the reviewer for raising calibration. Classical metrics such as Expected Calibration Error (ECE; [1]) are designed for classification settings where models expose explicit confidence scores for each class. They are ill-defined for generative ranking baselines like DG, which produce ordered lists via beam search rather than calibrated probability distributions over the cluster space. In contrast, the recommender-systems literature typically defines calibration as distributional alignment between a user’s interest profile and the recommended set [2]. Following this notion, we evaluate calibration via Jensen-Shannon Divergence (JS-Div), measuring how closely the predicted preference distribution matches the empirical next-window distribution. As reported in Table 2, PET consistently achieves the lowest JS-Div (e.g., 0.279 vs. 0.406 for the Qwen8B-reranker baseline), indicating that our inferred preference mass is substantially better aligned with users’ empirical behavior than competing methods.

---

> ### Author Response · Authors · 2025-12-03
>
> **Strengthening Theoretical Grounding: Learning Guarantees and Structural Analysis**
>
> We thank the reviewer for the rigorous critique regarding the theoretical substance. This feedback prompted us to formalize the statistical underpinnings of PET. In the revision, we establish that PET is grounded in statistical learning theory, moving beyond heuristic arguments to provide rigorous consistency results and generalization bounds. At a high level, Section 3.3 now summarizes the main theoretical story from an idealized “perfectly calibrated” regime (where the underlying preference distribution can be recovered) to an approximate, imperfect regime (where the learned distribution may incur error), and highlights the key implications for PET vs. direct generation – particularly why PET is expected to perform better on ranking quality/fairness and long-tail coverage. Appendix D contains the formal assumptions, propositions, and proofs underlying these claims, together with extended analysis of the approximate regime.
>
> - **Isotonicity is Derived, Not Assumed (Fisher Consistency).**
> We explicitly address the concern regarding circularity. In the revision (Appendix D.1), we demonstrate that strict isotonicity is not a heuristic assumption but the theoretical limit of the training objective. Leveraging the theory of Strictly Proper Scoring Rules [3], we prove that the global minimizer of the PET objective (Cross-Entropy) guarantees the recovery of the true preference distribution (Proposition D.1). This establishes a standard Fisher-Consistency result [4]: in the population limit, the correct ordering of logits is a mathematical consequence of the optimization process. This confirms the validity of the learning signal without requiring circular assumptions.
>
> - **Providing learning/generalization-style guarantees (the $\epsilon$–regret bound).**
> To address the request for generalization insight, we extended the analysis to the "imperfect" regime where the model is not at the global optimum. We introduce an $\epsilon$-approximate isotonicity condition and derive a bound on the pairwise ranking regret that scales as $O(K^2 \epsilon)$ (Theorem D.2), where $K$ is the number of clusters. Crucially, we bridge the gap between training and inference using Pinsker’s Inequality (Lemma D.1). We prove that minimizing the excess Cross-Entropy risk implicitly minimizes the upper bound on ranking errors. This provides the requested learning guarantee: it mathematically links the training objective to downstream ranking quality.
>
> - **Deepening Insight: Structural Analysis vs. Direct Generation.**
> We have reframed the theoretical contribution to focus on structural robustness. We moved the idealized optimality theorem to Appendix D as a limiting case. In the Main Text (Section 3.3), we now provide a Structural Analysis of why PET outperforms Direct Generation (DG). We mathematically identify three mechanisms where DG fails due to autoregressive pathology:
>    - (1) **Frequency Bias:** Pre-training popularity bias leaks into the ranking [5,6].
>   - (2) **Search Pathology:** Beam search imposes hard truncation on valid tail clusters [7,8].
>   - (3) **Exposure Bias:** Sequential error propagation distorts context [9,10].
>
>   PET, in contrast, reads out a full preference distribution in one shot from the user history, without sequential conditioning, search truncation, or next-token calibration artifacts.

---

> ### Author Response · Authors · 2025-12-03
>
> **Response on Computational Overhead (Likelihood-based Probing)**
>
> We respectfully point out a misconception regarding the computational cost of Likelihood-based Probing (LP), and disagree that LP is prohibitively expensive in our setting. The reviewer’s $O(K)$ characterization implicitly assumes large $K$ and full forward passes per cluster. In practice, LP reuses the KV-cache of the user’s history and decodes only a **constant-length probe snippet** per cluster (a few tokens for the template plus the cluster identifier). The total token-level cost per user is $L_{\text{ctx}} + K \cdot L_{\text{probe}}$,
>  where $L_{\text{ctx}}$ is the length (in tokens) of the user’s history context and $L_{\text{probe}}$ is a small constant. On MovieLens and Yelp, ($K \le 19$), so this amounts to one context encoding plus at most a few dozen additional tokens -- comparable to, and in many cases lighter than, Direct Generation, which must autoregressively decode an entire textual list of clusters (often tens of tokens) using beam search or top-$k$ sampling.
>
> Importantly, PET is not intended as an item-level reranker over millions of IDs: LP is used only in the cluster-level regime with small $K$. For large cluster spaces (e.g., Yelp), our framework explicitly switches to Generative Classification and Hierarchical Probing, which avoid $O(K)$ passes by scoring clusters in a single (or logarithmic) number of forward calls. Thus, in the regimes where we apply LP, its computational cost is modest and practical (no long-sequence generation), and for extreme $K$ we already provide PET variants that are computationally comparable to standard LLM-based recommenders.

---

> ### Author Response · Authors · 2025-12-03
>
> **On TikTok dataset details**
>
> We thank the reviewer for pointing that out and have included more details in our revised manuscript.
>
> **On reliance on predefined cluster taxonomies.**
>
> We thank the reviewer for raising this point and we acknowledge that the choice of cluster taxonomy is important, but PET is deliberately designed to be agnostic to how clusters are constructed. The framework only requires: (i) a mapping from items to clusters, and (ii) a human-readable description for each cluster. The LLM then reasons over these semantic descriptors; it does not rely on hard-coded IDs or task-specific label engineering. In fact, our experiments already cover heterogeneous taxonomies: MovieLens uses 19 fixed genres; Yelp uses a two-level business taxonomy whose level-1 mapping is generated by an external LLM agent (and never shown during PET SFT); TikTok uses a separate set of content themes. Across these very different cluster spaces, PET yields consistent gains in NDCG, long-tail NDCG, and JS-divergence, suggesting that performance is not brittle with respect to a particular hand-tuned taxonomy.
>
> Regarding granularity and automatic clustering, PET treats the cluster lattice as a modular input. Coarser taxonomies (e.g., MovieLens genres) give more compact, interpretable preference vectors; finer-grained taxonomies (e.g., Yelp L2 or TikTok subthemes) increase dimensionality but are naturally handled by our Generative Classification and hierarchical probing variants. In practice, the cluster space could be defined by any automatic or learned method -- e.g., clustering item embeddings, topic models, or behavior-based segments -- and then labeled with short textual descriptions. Because the LLM operates on these semantic descriptions, PET can ingest such automatically derived clusters without architectural changes. We will clarify this in the revised version and explicitly position the taxonomy construction as an orthogonal design choice: PET is a distributional user-modeling layer that can sit on top of either human-designed or learned cluster spaces.
>
> **Reference**
>
> [1] Chuan Guo, Geoff Pleiss, Yu Sun, and Kilian Q. Weinberger. "On Calibration of Modern Neural Networks." ICML 2017.
>
> [2] Harald Steck. "Calibrated recommendations." KDD 2018.
>
> [3] Gneiting, Tilmann, and Adrian E. Raftery. “Strictly Proper Scoring Rules, Prediction, and Estimation.” Journal of the American Statistical Association, 2007.
>
> [4] Tewari, Ambuj, and Peter L. Bartlett. “On the Consistency of Multiclass Classification Methods.” In COLT 2007: Conference on Learning Theory, 2007.
>
> [5] Zhao, Tony Z., Eric Wallace, Shi Feng, Dan Klein, and Sameer Singh. “Calibrate Before Use: Improving Few-Shot Performance of Language Models.” ICML 2021.
>
> [6] Holtzman, Ari, Jan Buys, Li Du, Maxwell Forbes, and Yejin Choi. “The Curious Case of Neural Text Degeneration.” ICLR 2020.
>
> [7] Stahlberg, Felix, and Bill Byrne. “On NMT Search Errors and Model Errors: Cat Got Your Tongue?” EMNLP 2019.
>
> [8] Wiseman, Sam, and Alexander M. Rush. “Sequence-to-Sequence Learning as Beam-Search Optimization.” EMNLP 2016.
>
> [9] Ranzato, Marc’Aurelio, Sumit Chopra, Michael Auli, and Wojciech Zaremba. “Sequence Level Training with Recurrent Neural Networks.” ICLR 2016.
>
> [10] Bengio, Samy, Oriol Vinyals, Navdeep Jaitly, and Noam Shazeer. “Scheduled Sampling for Sequence Prediction with Recurrent Neural Networks.” NeurIPS 2015.
>
> [11] Schick, Timo, and Hinrich Schütze. “It’s Not Just Size That Matters: Small Language Models Are Also Few-Shot Learners.” NAACL 2021.
>
> [12] Min, Sewon, Xinxi Lyu, Ari Holtzman, Mikel Artetxe, Mike Lewis, Hannaneh Hajishirzi, and Luke Zettlemoyer. “Rethinking the Role of Demonstrations: What Makes In-Context Learning Work?” EMNLP 2022.

---

### Official Review · Reviewer_X9UN · 2025-11-01

**Soundness:** 3
**Presentation:** 3
**Contribution:** 2
**Rating:** 4
**Confidence:** 3

**Summary:**

This paper argues that while traditional large language model (LLM)-based recommender systems can predict users' next actions, they lack interpretability and tend to amplify popular preferences while neglecting long-tail interests. To address this, the authors propose the PET framework, transforming the user modeling task from "generating recommendation lists" to "inferring the probability distribution of user preferences." Through logit probing and generative classification methods, the model can infer the probability distribution of user preferences across different preference categories from their history, thus achieving interpretable, traceable, and fair personalized recommendations. Theoretically, the method is proven to achieve optimal ranking results and solves the problems of multi-label and large-scale categories through a hierarchical inference mechanism. Overall, PET provides a transparent, scalable, and interpretable new paradigm for user modeling.

**Strengths:**

Highly interpretable: PET does not directly predict specific recommendations, but instead outputs the probability distribution of users across different interest categories. This allows the system to clearly demonstrate "why users like certain types of content," enabling transparent and traceable recommendation decisions.

Capturing Preference Changes: Through time-sliding windows and distribution evolution modeling, PET can dynamically track changes in user interests over time, distinguishing between short-term fluctuations and long-term trends, thus better understanding the evolution of user behavior.

Reducing Popularity Bias: Because it models the preference distribution at the category level rather than the popular items themselves, PET can balance mainstream and long-tail content in recommendations, preventing the system from overly favoring popular items and improving fairness and diversity.

**Weaknesses:**

Lack of comparability with traditional recommendation models:
PET outputs a preference distribution rather than a specific recommendation list, making it difficult to directly compare with traditional recommendation systems (such as MF, SASRec, and BERT4Rec) on metrics like click-through rate (CTR) or NDCG, thus hindering the quantifiable evaluation of its practical effectiveness.

High computational and training costs:
PET is based on a Large Language Model (LLM), requiring training and inference on large-scale corpora and long-sequence behavioral data. This results in significantly higher computational costs than traditional deep recommendation models, making it unsuitable for deployment in resource-constrained environments.

Strong dependence on reward signals and category segmentation:
The model's accuracy depends on the quality of preference cluster segmentation and the reliability of implicit feedback (such as clicks and views) in the training data. Inappropriate category segmentation or noisy feedback can distort the preference distribution, affecting tracking performance and interpretability.

The methods lack uniformity and do not discuss other possible paradigms for solving recommendation problems. For example, interpretable recommendation systems can also be achieved through the R1 reasoning paradigm. Related works, such as R1-ranker and Rec-R1, need to be discussed and compared.

**Questions:**

See weaknesses.

---

> ### Author Response · Authors · 2025-12-03
>
> **On "lack of comparability with traditional recommendation models"**
>
> We thank the reviewer for raising this point. This concern primarily comes from interpreting PET as an end-to-end item recommender, which is not the problem we formulate in the paper. PET is designed as a distributional user modeling framework: it infers a probability vector over semantically meaningful preference clusters, and tracks how that distribution evolves over time. In contrast, MF, SASRec, BERT4Rec, and similar models operate in the item-ID space and are optimized to output ranked lists over a given candidate set.
>
> Under our stated problem setting, the primary object of interest is the cluster-level preference distribution, not direct next-item CTR. As such, a one-to-one comparison to item-level recommenders on CTR/NDCG is not the central evaluation axis. Nevertheless, when we do instantiate PET as part of a ranking pipeline, we explicitly map PET’s distributions back to items and evaluate standard ranking metrics: we compare against (i) Direct Generation using the same backbone fine-tuned LLM and (ii) a strong public reranker (Qwen3-Reranker-8B), and on the industrial dataset we compare against a production Transformer ranker. These experiments show that PET yields competitive or better ranking quality in addition to its interpretability and temporal profiling benefits. We will make this positioning more explicit to avoid the impression that PET "cannot be compared" to traditional models; rather, PET tackles a different modeling target, and when plugged into ranking, we already provide item-level quantitative comparisons.
>
> **On "computational overhead of PET and Likelihood-based Probing"**
>
> We thank the reviewer for posing this question. And we respectfully disagree with the characterization of Likelihood-based Probing (LP) as prohibitively expensive in our setting. The $O(K)$ discussion is accurate only under a worst-case assumption of $K$ large full forward passes per cluster. In practice, LP reuses the KV-cache of the user’s history and decodes only a constant-length probe per cluster. The token-level cost per user is $L_{\text{ctx}} + K  \cdot L_{\text{probe}}$ where $L_{\text{ctx}}$ is the length (in tokens) of the user’s history context and $L_{\text{probe}}$ is a small constant (on MovieLens/Yelp this is only a few tokens per cluster). On a small cluster space like MovieLens, where $K \leq 19$, this amounts to one context encoding plus at most a few dozen additional tokens -- comparable to, and often lighter than, Direct Generation, which must autoregressively decode an entire textual list of clusters (typically tens of tokens) with beam search or top-$k$ sampling.
>
> Moreover, PET is not applied as an item-level reranker over millions of IDs. LP is used only in the cluster-level regime with small $K$. For large cluster spaces, our framework explicitly switches to Generative Classification and Hierarchical Probing, which avoid $O(K)$ decoding by scoring clusters in a single (or logarithmic) number of forward calls. Thus, in the regimes where we use LP, the computational cost is modest and practical, and for extreme $K$, we already provide PET variants that are computationally comparable to other LLM-based recommenders. We will clarify these distinctions so that the complexity discussion matches the actual usage regimes of each PET variant.

---

> ### Author Response · Authors · 2025-12-03
>
> **On "dependence on reward signals and category segmentation"**
>
> We thank the reviewer for this point, and we recognize that interaction signals and category definitions matter, but this is not a PET-specific fragility; it is inherent to all data-driven preference-learning and recommendation systems. Implicit feedback (clicks, views, play duration) is widely recognized as noisy and ambiguous, and large-scale systems often operate with incomplete or inconsistent content taxonomies. This challenge is documented both in industrial systems such as YouTube, which explicitly model noisy implicit feedback and note the lack of a well-defined content ontology, and in surveys and recent work on denoising implicit-feedback recommenders [1,2].
>
> Within this general landscape, PET is designed to be agnostic to how the empirical label distribution is constructed: in our experiments we adopt simple, transparent definitions (e.g., normalized interaction counts or play duration), but alternative constructions can be plugged in without changing the architecture. Importantly, PET can actually be more robust than ID-based models because it operates at the level of semantic clusters. Cluster names and descriptions are natural language, allowing the LLM to align semantically related labels (e.g., “Sci-Fi” vs “Science Fiction”) even when the underlying ontology is imperfect. Aggregating item-level interactions into clusters and normalizing to a probability vector also smooths spurious noise from individual items. We will emphasize in the revision that PET does not introduce a new dependence on noisy feedback or taxonomies beyond what is standard in recommender systems, and, in fact, its semantic, cluster-level formulation mitigates some of these issues.
>
> **On “methods lack uniformity” and "missing discussion of R1-style paradigms"**
>
> We thank the reviewer for listing the references. Methodologically, the PET variants are unified: Likelihood-based Probing, Generative Classification, and Hierarchical Probing all share the same core objective -- reading out a cluster-level preference distribution from LLM logits. They differ only in the computational regime they target (small vs. extreme $K$, accuracy vs. latency trade-offs), and thus represent implementation choices within a single framework rather than unrelated methods. We will make this shared objective more explicit to avoid the impression of fragmentation.
>
> Regarding R1-style paradigms (R1-ranker, Rec-R1, etc.), these works address a different level of the recommendation problem. They perform item-level ranking via chain-of-thought reasoning and textual explanations, whereas PET intentionally models users at the cluster-distribution level. Our goal is to obtain a compact, interpretable probability vector over preference clusters that can be reused across tasks (profiling, fairness analysis, long-tail coverage), rather than to generate textual rationales for each ranked item. In this sense, PET and R1-style rankers are complementary, not competing: PET can provide a structured user preference prior that an R1-style ranker conditions on. We will expand the related-work discussion to clarify this distinction and to position R1-style models as complementary item-level reasoning approaches rather than direct baselines for our distributional user modeling task.
>
> [1] Covington, et al. Deep neural networks for youtube recommendations.
>
> [2] Jannach, et al. Recommending based on implicit feedback.

---

### Note · Authors · 2026-01-07

I have read and agree with the venue's withdrawal policy on behalf of myself and my co-authors.